# RecA finds homologous DNA by reduced dimensionality search

Jakub Wiktor[1,3], Arvid H. Gynnå[1,3], Prune Leroy[1], Jimmy Larsson[1], Giovanna Coceano[2], Ilaria Testa[2] & Johan Elf[1✉]

Homologous recombination is essential for the accurate repair of double-stranded DNA breaks (DSBs)[1]. Initially, the RecBCD complex[2] resects the ends of the DSB into 3′ single-stranded DNA on which a RecA filament assembles[3]. Next, the filament locates the homologous repair template on the sister chromosome[4]. Here we directly visualize the repair of DSBs in single cells, using high-throughput microfluidics and fluorescence microscopy. We find that, in *Escherichia coli*, repair of DSBs between segregated sister loci is completed in 15 ± 5 min (mean ± s.d.) with minimal fitness loss. We further show that the search takes less than 9 ± 3 min (mean ± s.d) and is mediated by a thin, highly dynamic RecA filament that stretches throughout the cell. We propose that the architecture of the RecA filament effectively reduces search dimensionality. This model predicts a search time that is consistent with our measurement and is corroborated by the observation that the search time does not depend on the length of the cell or the amount of DNA. Given the abundance of RecA homologues[5], we believe this model to be widely conserved across living organisms.

To study the mechanism of homologous recombination in living bacteria, we created an inducible DSB system consisting of (1) an inducible Cas9 nuclease to create DSBs at a specific chromosomal location (the 'cut site'), (2) a fluorescent *parS*/mCherry–ParB system (hereafter referred to as ParB) to visualize the chromosomal location of the break, and (3) an SOS-response[6] reporter to select the cells undergoing DSB repair (Fig. 1a, Extended Data Fig 1a). We used a variant of the microfluidic mother machine[7] that allows for brief induction of the Cas9 nuclease (Fig. 1a, Extended Data Fig. 1b).

### DSB repair is fast and accurate

Formation of a DSB is followed by end processing by RecBCD, which removes ParB markers close to the break[8], and by activation of the SOS-response reporter (Fig. 1b). A pulse of Cas9 induces specific DSBs in cells with the chromosomal cut site, which is accompanied by an increase in the fraction of SOS-activated cells shortly after the induction (Fig. 1c, d, Extended Data Fig. 2a). Expression of Cas9 in cells without the cut site or expression of catalytically dead Cas9 (dCas9) did not induce an SOS response (Fig. 1d, Extended Data Fig. 2b). Activation of the SOS response depended on homologous recombination and was absent in cells lacking *recA* or *recB* genes (Extended Data Fig. 2b). The ParB foci lost owing to DSBs were recovered and segregated in wild-type cells, but not in cells lacking critical components of recombination (Extended Data Fig. 2c, f). These results show that the induced DSBs were repaired by homologous recombination. The repair is notably robust: 95.5% (447 out of 468) of cells that retained an uncleaved template repaired the DSB and subsequently divided. Repair was impaired in cells without a repair template, as none of the cells that cleaved all copies of the cut site divided during 4-h experiments (*n* = 27 cells, 3 experiments).

Next, we focused on the dynamics of the cut site loci during repair. Typically, after a DSB, the uncut locus first translocated to the middle of the cell and then split into two foci that subsequently segregated (Fig. 1c). This pattern was visible when we plotted the positions of the ParB foci along the cell length against the time relative to the DSB event (Fig. 1e). We then measured repair times in individual cells, that is, the time between the loss and reappearance of the ParB focus (Fig. 1c). We limited the analysis to cells that had two separated ParB foci before the DSB. We found that a DSB is repaired in 15.2 ± 5.0 min (Fig. 1g). These results were consistent between replicates (Extended Data Fig. 2d), and notably, when I-SceI was used instead of Cas9 to induce breaks (Extended Data Fig. 2e), or when the ParB marker was replaced by a *malO* array bound by MalI–Venus proteins (referred to as 'MalI') (Extended Data Fig. 3a). When the DSB was flanked by ParB on one side and MalI on the other, both ends were processed simultaneously and followed the same dynamics (Extended Data Fig. 3b). Given that the repair time is just a fraction of the generation time (here 35 ± 10 min), we tested whether it has negative effects on fitness. A single DSB delayed the division to 55 ± 10 min; however, this delay was compensated by faster divisions in the following generations (Fig. 1f). The growth rate was temporarily reduced by 4% in cells undergoing DSB repair (Fig. 1f).

### Pairing between distant homologies

The loss of ParB foci prevented observation of the cleaved cut site locus. Therefore, to visualize the dynamics of the break site, we used a set of MalI markers integrated at distances that are not processed by RecBCD[8,9] (−25 kb or +170 kb from the cut site) (Fig. 2b). Imaging either of the two MalI markers showed that after a DSB, both sister

[1]Department of Cell and Molecular Biology, Science for Life Laboratory, Uppsala University, Uppsala, Sweden. [2]Department of Applied Physics, Science for Life Laboratory, KTH Royal Institute of Technology, Stockholm, Sweden. [3]These authors contributed equally: Jakub Wiktor, Arvid H. Gynnå. ✉e-mail: Johan.elf@icm.uu.se

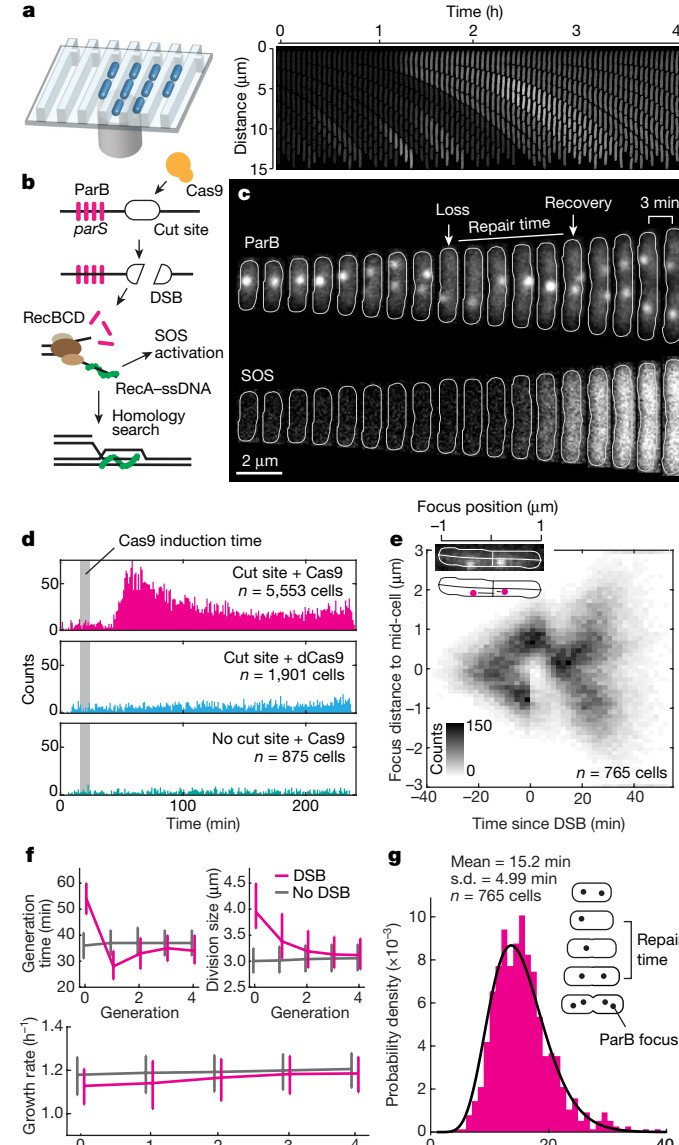

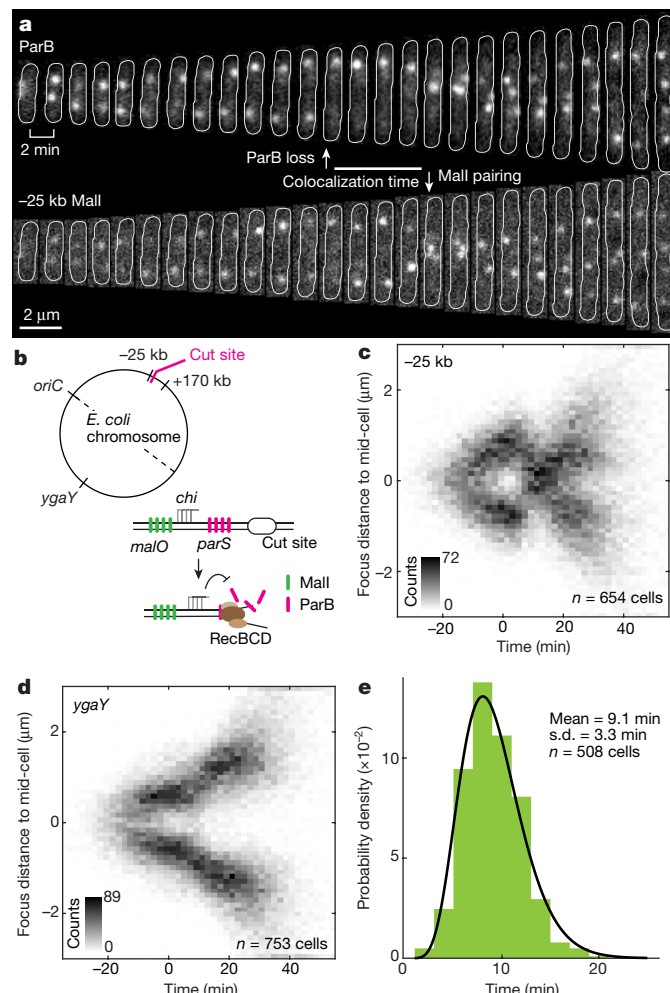

**Fig. 1 | High-throughput imaging of DSB repair. a**, Left, cartoon showing a mother machine device used to grow cells. Right, montage of a single growth channel showing SOS activation after induction of DSBs. **b**, Cartoon showing formation, processing and repair of a DSB. Cas9 binds to the cut site and creates a DSB. Next, RecBCD binds to DNA ends at the DSB site and begins end processing, ejecting ParB proteins and generating a 3′ ssDNA tail to which RecA binds and induces the SOS response. The RecA–ssDNA filament searches for homology, and after homology is located the DSB is repaired. **c**, A cell undergoing DSB repair. Loss and recovery of ParB focus are annotated. Cell outlines are displayed with white lines. **d**, The time of SOS induction in individual cells, defined as the time from the start of the Cas9 induction to the time at which the CFP signal from the SOS reporter reached half of the maximum signal. Only cells that increased CFP signal by more than fourfold were plotted. Data were aggregated from four (top) or three (middle, bottom) experiments. **e**, Localization of ParB foci along the cell length during DSB repair. Cells were oriented so that the remaining ParB focus was positioned on top. Time was aligned by ParB focus loss, as annotated in **c**. Insert shows a ParB channel of a single cell overlaid with outline and backbone (top), and mapping of the foci position along the cell's backbone (bottom), $n = 717$ cells from 4 experiments. **f**, Generation time (top left), size at the division (top right) and growth rates (bottom) of cells undergoing repair of a single DSB and their descendants. The reference (no DSB) sample consists of the cells without a DSB that were born during the time window of maximal DSB induction. The line represents the median and error bars show the first and the third quartile, $n = 60,405$ cells from 4 experiments. **g**, DSB repair times, with gamma distribution fitted to the data, $n = 765$ cells from 4 experiments.

**Fig. 2 | Pairing between segregated sisters. a**, Cell with −25-kb MalI marker undergoing DSB repair. ParB focus loss and colocalization of MalI foci are annotated. Cell outlines are displayed with a white line. **b**, Top, cartoon showing the circular *E. coli* chromosome with the locations of inserted cut site and *malO*/MalI markers. Bottom, cartoon showing the processing of a DSB in presence of the *malO* array protected by *chi* sites. The *chi* sites prevent RecBCD from degrading the DNA before reaching *malO* sites. **c**, Localization of −25-kb MalI foci in cells undergoing DSB. For each cell, the time was aligned by ParB focus loss. $n = 654$ cells from 3 experiments. **d**, As **c**, but for the *ygaY* MalI marker, on the opposite chromosomal arm to the cut site. $n = 753$ cells from 2 experiments. **e**, Distribution of MalI foci colocalization times for −25 kb MalI marker during DSB repair. The solid line shows the gamma fit to the data. $n = 508$ cells from 3 experiments.

loci moved to the middle of the cell, where they colocalized (Fig. 2a, c, Extended Data Fig. 4a, c). The movement of the sister MalI markers was symmetric, unlike previously reported dynamics in which the cleaved sister locus moved much further after a DSB[10,11]. To test whether the colocalization of sister loci is specific or caused by a global alignment of the chromosomes, we used a distant MalI marker integrated on the opposite arm of the chromosome (*ygaY*). The distant *ygaY* marker maintained its typical position during repair (Fig. 2d, Extended Data Fig. 4b), excluding a model in which homologous recombination repair induces pairing of entire chromosomes. Instead, it appears colocalization is specific for the cleaved cut site and its homology. As the sister locus is no different from any other chromosomal locus until it has been located through search, we concluded that the colocalization of sister loci implies completed homology search (Extended Data Fig. 4e). The −25-kb markers colocalized 9.1 ± 3.3 min (Fig. 2e) after the DSB, and thus the homology search is faster than this.

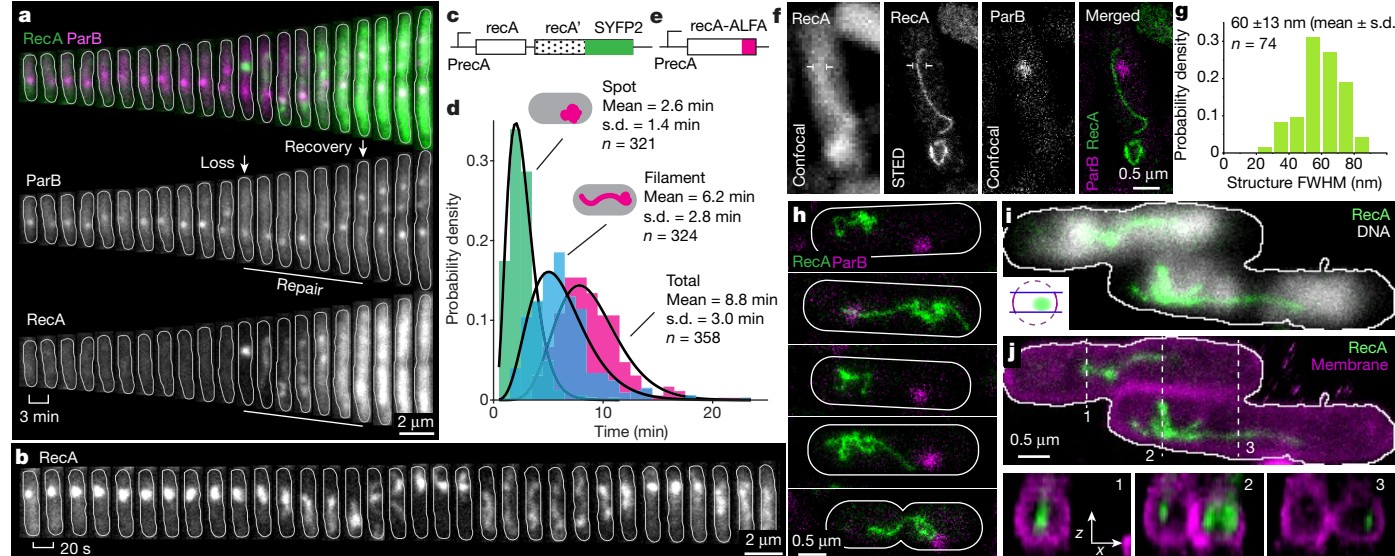

**Fig. 3 | Thin RecA filaments mediate search for homology. a**, RecA–YFP (green) and ParB (magenta) during DSB repair. Cell outlines are shown with a white line. **b**, Fast time-lapse of RecA–YFP during DSB repair. **c**, RecA–RecA–YFP tandem construct inserted into the *recA* locus. **d**, Lifetimes of RecA structures. The solid line shows the gamma fit to the data. *n* = 3 experiments, numbers of analysed cells are shown. **e**, RecA–ALFA construct inserted in the *recA* locus. **f**, Cell with an extended RecA–ALFA structure imaged in confocal and STED microscopy. Intensity profile between white markers is shown in Extended

Data Fig. 6e. **g**, Distribution of thickness of RecA–ssDNA filaments measured as indicated in Extended Data Fig. 6e. *n* = 74 from 2 experiments. **h**, STED images of cells with RecA–ALFA structures. White lines indicate cell outlines. **i**, Three-dimensional STED image of RecA–ALFA overlaid with confocal image of DNA stained by Pico488. Only the middle 400-nm section is shown (inset). **j**, Top, 3D STED image of RecA–ALFA overlaid with cell membrane stained with Nile red. Bottom, cross-sections at highlighted locations.

## RecA filaments are thin and dynamic

The homology search is mediated by a RecA filament assembled on single-stranded DNA (ssDNA)[3] and structures made by fluorescent RecA fusions have previously been imaged[10–14]. We visualized RecA activity during DSB repair using a RecA–yellow fluorescent protein (YFP) fusion integrated in tandem with wild-type *recA*, a construct found to be fully active (Fig. 3c, Extended Data Fig. 5b, g). A similar construct has previously been shown to be fully functional[14] but has not been used to characterize distant repair events.

Induction of DSBs led to the *recB*-dependent formation of RecA structures at the DSB sites (Fig. 3a, Extended Data Fig. 5a, f). These structures appeared 35 ± 98 s (Extended Data Fig. 5h) before the loss of the ParB focus, and disassembled 6.6 ± 5.2 min (Extended Data Fig. 5i) before repair was complete, as defined by segregation of the ParB foci. The SOS response is activated by RecA filament assembled on ssDNA[3]. We predicted that if the structures are RecA–ssDNA filaments, their lifetime would correlate with the strength of the SOS response. As this was indeed the case (Pearson's *r* = 0.36, *P* = 3 × 10[−12]) (Extended Data Fig. 5d, e), we are confident that the structures are RecA–ssDNA filaments.

The RecA structures were highly dynamic on a timescale of tens of seconds (Fig. 3b). Their average lifetime was 8.8 ± 3.0 min (Fig. 3d), and during that time they displayed two forms: the initial spot at the DSB (existing for 2.6 ± 1.4 min) that increased in intensity (Extended Data Fig. 5a), suggesting RecA loading on ssDNA; and a filament extruding from the initial spot and extending throughout the cell (existing for 6.2 ± 2.8 min).

It has been reported that RecA forms bundle-like structures during bacterial DSB repair[10]. To test whether we could observe such structures, we used stimulated emission depletion super-resolution microscopy (STED) to image RecA tagged with the ALFA epitope[15]. RecA–ALFA alone was fully functional in DSB repair (Extended Data Fig. 5b, g). Immunostaining of cells undergoing DSB repair revealed that RecA forms thin, filamentous structures (Fig. 3f, h, Extended Data Fig. 6a) that were not associated with the membrane—rather, they appeared in the central region of the cell (Fig. 3i, j, Extended Data Fig. 6b–d, Supplementary

Video 1). Deconvolution of the observed 60 ± 13 nm full width at half maximum (FWHM) with the point spread function (PFS) of the imaging system showed that the filament width is 37.5 ± 23.5 nm (Fig. 3g, Extended Data Fig. 6e, f). As expected, given the high mobility of the RecA–YFP structures in living cells, fixed cells exhibited a large variety of conformations, including complex, tangled threads or single, winding filaments spanning the length of the cell (Fig. 3h). Notably, we observed the same type of structures when immunostaining RecA–YFP in the tandem construct (Extended Data Fig. 6g), or the RecA–ALFA, RecA–YFP tandem construct (Extended Data Fig. 6h–k). These results show that the homology search is performed by thin and flexible RecA filaments.

## RecA reduces dimensionality of search

The fast target search by RecA compared with other systems that also rely on homology-directed search[16] requires a different quantitative model. Owing to the slow diffusion rates of both the large RecA–ssDNA complex and the chromosomal loci[17], the search cannot be explained by bi-molecular reaction–diffusion in three dimensions[18]. We propose that the RecA homology search is facilitated by a 'reduced dimensionality' mechanism that accelerates the process in two ways. First, ATP hydrolysis enables mechanical extension of the RecA–ssDNA filament across the cell in less than a minute, rapidly covering most of the distance between the broken ends and the search target as shown in Fig. 3a. Second, the extended filament interacts with many different sequences in parallel. Such simultaneous probing has previously been suggested on the basis of single-molecule experiments[19] and cryo-electron microscopy structures[20]. Our addition to the model is the realization that at any *z* coordinate (that is, along the long axis of the cell) there is always at least one segment of the stretched RecA–ssDNA filament that is homologous to a double-stranded DNA (dsDNA) segment at the same *z* position (Fig. 4a). This makes the search problem independent of the *z* coordinate and reduces the complexity from three to two dimensions. That is, the time of homology pairing is equal to the time it takes for a segment of chromosomal dsDNA to diffuse

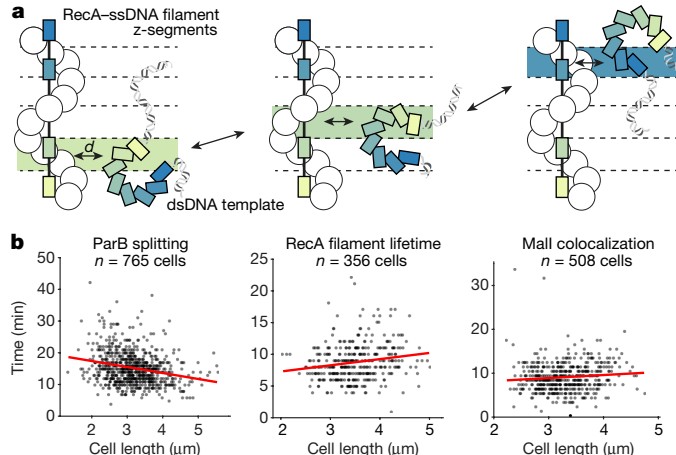

**Fig. 4 | Reduction of dimensionality explains fast search times. a**, The RecA–ssDNA filament and the repair template share one homologous segment (coloured bars) at each $z$ coordinate along the length of a cell. As the repair template moves along the cell, only the segment at the current $z$ coordinate is relevant. The search by the relevant segment thus occurs in 2D. **b**, Repair times as a function of cell length at the time of the break. The times were measured based on ParB foci splitting (left), RecA filament lifetime (middle) and MalI colocalization (right). Red lines display linear fit to the data. Numbers of analysed cells are displayed, $n = 4$ (ParB) or 3 (RecA, MalI) experiments.

radially to the RecA–ssDNA filament, and not the time it takes for two segments to find each other by 3D diffusion in the whole cell. In this case, 2D search is approximately 100 times faster than 3D search[21] (for a detailed description, see Methods).

In the Supplementary Information, we derive an expression for the expected time for the homologous sequences to encounter each other. Our model predicts that the search will be completed within 5 min on the basis of the rate of diffusion of the target DNA at the length scale of the radius of the nucleoid[17]. Of note, the model also predicts that the search time should be invariant with respect to the cell length, as only the radial distance is relevant. This distinguishes our model from previous models, such as the one presented in ref. [22]. Experimental data confirm that the length of a cell has a minuscule effect on the search time estimated by either ParB foci splitting, RecA structure lifetimes or MalI foci colocalization in the mid-cell (Fig. 4b), despite larger cells having more DNA to explore (Extended Data Fig. 8).

## Discussion

We propose that the stretched RecA–ssDNA filament—in a simple and elegant way—positions at least one ssDNA segment in the proximity of its homologous sister, such that the homologous dsDNA segment can find the ssDNA segment using a fast, short-range search. RecA is a prototypic member of the strand-exchange protein family, which is found in all forms of life and shares a common mechanism[3,5]. A reduced-dimensionality mechanism may be a conserved property of these proteins. The advantage of the stretched filament is obvious in elongated cells, and long RecA structures have indeed been observed in *Caulobacter crescentus*[11] and *Bacillus subtilis*[13]. When DSB repair occurs at the replication fork, where the sister template is nearby, fluorescent RecA forms only a transient focus at the site of the break[14], suggesting that the search may finish before the RecA filament is fully extended. Previous work has suggested that the repair of DSBs induced by I-SceI is carried out by 'bundles', which are complex structures formed by RecA[10]. These bundles are characterized by a thick central body, low mobility, tens-of-minutes lifetime, and are positioned between the nucleoid and the inner membrane. We have found that the RecA filaments involved in DSB repair are markedly different from these bundles:

they are thin, dynamic, last only for minutes, and are present within the nucleoid (Fig. 3h, i). Notably, according to the reduced-dimensionality model, the search time is not affected by an increase in the amount of DNA, as long as the length of the filament scales with the amount of DNA. The mechanism therefore enables search in organisms with genomes larger than those of bacteria.

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

## Methods

### Strain construction

Strains used in this work are derivatives of *E. coli* TB28[23] in which we restored the *rph-1* mutation to wild type and deleted the *malI* gene and *malO* operator. Genetic integrations were done using lambda red integration[24], resistance cassettes were removed using the pCP20 plasmid[24], and markers were combined using P1 phage transduction.

The labels mCherry-ParBMt1[25] and MalI-Venus were expressed by constitutive promoters integrated into the *gtrA* locus. The *malO*/MalI marker consists of an array of 12 maltose operators, which are the binding sites for MalI-Venus.

The DSB cassette consists of the I-SceI recognition site flanked by two *lac* operators, a *parSMt1* site and three *chi* sites positioned outside the parSMt1 and I-SceI recognition sequence (Extended Data Fig. 1). The Cas9 target site was chosen about 100 bp away from the *parSMt1* site. Notably, the construct is designed in a way that there are no *chi* sites between the DSB site and *parSMt1* site. The DSB cassette was integrated into *codA* locus.

The RecA–YFP fusion was expressed directly downstream from the endogenous *recA* gene, and made by replacing mCherry with SYFP2 in the construct by ref. [14]. The RecA–ALFA fusions were made by introducing ALFA C-terminal to *recA* in either the wild-type *recA* locus or into the tandem construct mentioned above.

List of the strains used in this study can be found in Extended Data Table 1.

### Plasmid construction

p15a-SceIdeg-amp was cloned using HiFi DNA Assembly (NEB) by fusing two PCR fragments: (1) pSC101SceI_deg_amp[8] fragment amplified with Jwu035 and Jwu036, and (2) p15aSceI_deg_Kan[8] fragment amplified with Jwu037 and Jwu038. p15a-dSceIdeg-amp was cloned with Gibson assembly from two PCR fragments amplified from p15a-SceIdeg-amp template with two primer pairs: (1) Jwu088 and Jwu090, and (2) Jwu085 and Jwu091. p15a-Cas9deg-amp was cloned using Gibson assembly from two fragments: (1) p15a-SceIdeg-amp fragment amplified with Jwu273 and Jwu274, and (2) pCRED (gift from Daniel Camsund) fragment amplified with Jwu263 and Jwu264. Plasmid p15a-SceIdeg-amp-SOS was cloned using Gibson assembly protocol with: (1) p15a-SceIdeg-amp fragment amplified with primers Jwu330 and Jwu331, (2) fragment amplified form *E. coli* genome (from the strain EL1171) with primers Jwu332 and Jwu333. p15a-Cas9deg-amp-SOS plasmid was cloned using Gibson assembly with (1) p15a-wtCas9deg-amp fragment amplified with primers Jwu263 and Jwu272 and (2) p15a-SceIdeg-smp-SOS fragment amplified with primers Jwu273 and Jwu274. p15a-dCas9deg-amp-SOS was cloned using Gibson assembly protocol using (1) fragment amplified from p15a-SceIdeg-amp-SOS using primers Jwu273 and Jwu274 and (2) fragment amplified from *E. coli* genome (from the strain EL1605) using primers Jwu264 and Jwu272. pKD 13-recA::ALFA-recAsh-SYFP2-frt-kan-frt was cloned using golden gate protocol with BbsI restriction enzyme from (1) fragment amplified from *E. coli* genome (from the strain EL2515) with primers Jwu485 and Jwu486, (2) fragment amplified from *E. coli* genome (from the strain EL2699) using primers Jwu487 and Jwu488, and (3) fragment amplified from pKD13-P58-SYFP2-frt-kan-frt with primers Jwu489 and Jwu490.

psgRNA-CS1 was cloned in *E. coli* Top10 by blunt-end ligation of a fragment generated with a PCR with Jwu267 and Jwu184 and psgRNA[26] plasmid as a template (psgRNA was a gift from D. Bikard; Addgene plasmid no. 114005), the guide RNA (gRNA) sequence is ACTGGCTAAT GCACCCAGTA.

A list of primers used in this study can be found in Extended Data Table 2.

### High-throughput DSB imaging

**Growth conditions.** For the microfluidic experiments cells were grown at 37 °C in M9 medium supplemented with 0.4% glucose, 0.08% RPMI 1640 amino acids (Sigma-Aldrich R7131), surfactant Pluronic F-109 (Sigma-Aldrich 542342, 21 µg ml⁻¹), and when relevant carbenicillin (40 µg ml⁻¹) or kanamycin (20 µg ml⁻¹). Cells from −80 °C freezer stock were used to inoculate LB medium supplemented with adequate antibiotics and grown overnight at 37 °C. On the next day, the cells were diluted 1/250 in M9 0.4% glucose 0.08% RPMI medium and grown for 2 h, then cells were loaded onto the microfluidic chip. Cells were grown in the microfluidic chip for at least 2 h before the start of the experiments. Induction of Cas9 and dCas9 was done with a 5-min (or 6-min for spot counting experiments in Extended Data Fig. 2c) pulse of aTc (20 pg ml⁻¹).

**Microscopy.** Microscopy experiments were performed using a Ti2-E (Nikon) inverted microscope equipped with CFI Plan Apochromat DM Lambda 100× objective (Nikon), Sona 4.2B-11 sCMOS camera (Andor), and Spectra III (Lumencor) fluorescent light source. The microscope was controlled by Micro-Manager[27] running in-house build plugins. Fluorescence light source was triggered by the camera with the TTL connection. Custom fluorescent cubes were used: CFP excitation filter: BrightLine FF02-438/24 (Semrock), emission filter: BrightLine FF01-494/41 (Semrock), dichroic mirror: Di02-R442 (Semrock); YFP excitation filter: FF01-514/3-25 (Semrock), emission filter: ET550/50M 200362 (Chroma), dichroic mirror: Di02-R514 (Semrock); mCherry excitation filter: FF01-559/34 (Semrock), emission filter: T590LP 262848 (Chroma), dichroic mirror: T585lpxr (Chroma). Imaging was done with a 1.5× intermediate magnification lens. Phase-contrast images were taken with the CFP cube inserted. Typically, a phase-contrast image was acquired every minute with 80 ms exposure, CFP channel every third minute with fluorescence light intensity set at 5% and exposure of 40 ms, mCherry channel was acquired every minute with fluorescent light intensity set at 20% and exposure of 80 ms, YFP channel was acquired every minute (for imaging RecA), or every second minute (for MalI experiments) with fluorescence light intensity set at 40% and exposure of 100 ms. For spot counting experiments in Extended Data Fig. 2c the mCherry channel was imaged every third minute, and 3 *z* slices separated by 300 nm were taken at every time point and fluorescent images were reconstructed using maximum intensity projection.

**Microfluidic experiments.** Microfluidics experiments were performed using PDMS mother machine microfluidic chips developed previously[7]. This chip design allows for loading of two different strains and for automated switching of the medium. Medium pressure was controlled using an OB1 MK3+ microfluidic flow controller (Elveflow). The aTc was pulsed at the beginning of the for 3-h experiments, or 20 min after the start of image acquisition for 4-h experiments.

**Chromosome staining with DAPI.** Strains EL2504 and EL1743 (with *dnaC2*[28]) were grown overnight in LB medium. The next day, cell cultures were diluted in 5 ml of M9 with 0.4% glucose and 0.08% RPMI 1640 amino acids (Sigma-Aldrich R7131), and incubated at 37 °C (strain EL2504). The EL1734 strain was grown at 30 °C for 2 h before imaging one aliquot of the culture was grown at 42 °C to induce replication arrest. DAPI was added to a final concentration of 1 µg ml⁻¹, and cells with DAPI were incubated for 30 min at the growth temperatures. Then 1 ml of cells was centrifuged at 4 °C, 7,000 rpm (5424 R, Eppendorf) for 3 min and resuspended in 50 µl of cold ITDE (Integrated DNA Technologies) buffer supplemented with 10 mM MgCl₂. Two µl of concentrated cells were mounted on an agarose pad for imaging. Imaging was done with a 445-nm laser at the power 12 mW cm⁻² and exposure time of 220 ms. Phase-contrast images were segmented using Nested-Unet neural network[29], trained in-house. DAPI images were corrected for background by subtracting the mean pixel intensity of the area outside of the cell.

**Image analysis.** Data analysis was done in MATLAB (Mathworks), with the exception of the cell segmentation, which was done in Python. Microscopy data were processed using an automated analysis pipeline

developed previously[30], with several modifications. First, segmentation of phase-contrast images was done using Nested-Unet neural network[29] trained in-house, specifically for our microscopy setup. Pytorch 1.7.1 was used for the neural networks. We trained two networks, one to segment cells, and another to detect microfluidic growth channels on the phase-contrast images (Extended Data Fig. 1c). Transformation matrices between images acquired with different filter cubes were measured and fluorescent images were transformed to correct for the pixel shifts between fluorescence and phase-contrast images. Gramm[31] toolbox was used to generate some of the plots in Matlab.

**Selection and analysis of cells undergoing DSB.** Cells undergoing a DSB repair were selected on the basis of an increase of at least fourfold in CFP signal from plasmid-borne SOS reporter. First, the CFP image was background corrected by subtracting an image filtered with a Gaussian filter with the kernel size of 20 pixels (using the Matlab function imgaussfilt) from the original CFP image. We limited the analysis to the cells that had no major errors in segmentation, lived for at least 9 min, and divided during the experiment. Manual repair dynamics annotation was done only on cells that contained two ParB foci prior to the DSB, and divided after the repair. Cells that had >2 ParB foci, or that induced a DSB more than once, were excluded from the analysis. In the experiment in Extended Data Fig. 2c ParB foci were detected automatically using a radial symmetry-based method. Spot positions were mapped on the cell length using a custom written Matlab code. All images showing example cells come from the experiments that were repeated at least twice, with exception of costaining of RecA–YFP and RecA–ALFA, which was done once (Extended Data Fig. 6 h–k).

## Super-resolution STED microscopy
**Sample preparation.** Cells expressing RecA–ALFA, RecA–YFP, or both were grown for 3 h at 37 °C in M9 medium with glucose (0.4%), RPMI 1640 amino acids (0.08%), carbenicillin (20 µg ml[−1]) and kanamycin (10 µg ml[−1]). Cas9 was induced for 40 min with aTc (0.8 pg ml[−1]), after which the cells were fixed with formaldehyde (3.5%) for 10 min. Fixing was quenched with 100 mM glycine and the cells were washed in PBS before permeabilization in 70% ethanol for 1 h. For staining, the cells were blocked in PBS with BSA (1%) for 30 min and then incubated with antibodies at 1:200 dilution for at least 1 h. We used camelid single domain antibodies conjugated to either Star635P or Star580, for RecA–ALFA FluoTag-X2 anti-ALFA (N1502-Ab635P) and for RecA–YFP FluoTag-X4 anti-GFP (N0304-Ab635P and N0304-Ab580, NanoTag Biotechnologies). In the absence of epitopes, no antibody binding was detected (Extended Data Fig. 7). To visualize the nucleoid and membranes, the cells were stained with Pico488 (1:400 dilution, Lumiprobe) and Nile red (5 µM), respectively. After three washes in PBS, the cells were mounted on an agarose pad for microscopy.

**2D and 3D STED microscopy.** Super-resolution imaging was performed with a custom-built STED setup[32]. Excitation of the dyes was done with pulsed diode lasers; at 561 nm (PDL561, Abberior Instruments), 640 nm (LDH-D-C-640, PicoQuant) and 510 nm (LDF-D-C-510, PicoQuant). A laser at 775 nm (KATANA 08 HP, OneFive) was used as the depletion beam, which was split into two orthogonally polarized beams that were separately shaped to a donut and a top-hat respectively in the focal plane using a spatial light modulator (LCOS-SLM X10468-02, Hamamatsu Photonics), enabling both 2D- and 3D-STED imaging. The laser beams were focused onto the sample using a HC PL APO 100×/1.40 Oil STED White objective (15506378, Leica Microsystems), through which the fluorescence signal was also collected. The imaging was done with a 561-nm excitation laser power of 8–20 µW, a 640 nm excitation laser power of 4–10 µW and a 775 nm depletion laser power of 128 mW, measured at the first conjugate back focal plane of the objective.

Two-color STED imaging of RecA–YFP together with RecA–ALFA was done in a line-by-line scanning modality, averaging over 4 or 8 lines; while ParB and RecA–ALFA was recorded frame by frame, with the first channel in confocal and the second channel in STED. The pixel size for all 2D images was set to 20 nm with a pixel dwell time of 50 µs.

Volumetric two-color 3D-STED imaging of RecA-ALFA together with Nile red was recorded in a line-by-line scanning modality, while a single confocal frame of Pico488 was recorded at the middle of the bacterial cell afterwards. The voxel size for *xyz* volumes was set to 25 × 25 × 80 nm[3]. The pixel dwell time was set at either 30 or 50 µs.

Raw images were processed and visualized using the ImSpector software (Max-Planck Innovation) and ImageJ[33,34]. Brightness and contrast were linearly adjusted for the entire images. The size of the RecA filaments and RecA bundles were calculated tracing line profiles perpendicular to the structure orientation and averaged on two pixels on the raw images. The data were then fitted with a Gaussian function with the software OriginPro2020, from which the full width half maximum was extracted. For *xz* representation, images were deconvolved using the Richardson Lucy deconvolution algorithm implemented in ImSpector. 3D volumetric rendering was done with Huygens deconvolution in Imaris 9.1 (Bitplane).

The resolution of the microscope was measured on a calibration sample, made of sparse antibodies attached to the glass, coupled with the Star635P dye. The line profiles were extracted and fitted with a Lorentzian function[35], from which the width ($W$) was extracted as the dot size.

We estimated the diameter of the filaments to be 37.5 ± 23.5 nm as a deconvolution of observed 60 ± 13 nm FWHM width (Fig. 3g) with the 35 ± 11 nm (FWHM) Lorentzian PFS of the imaging system (Extended Data Fig. 6f) assuming that filament is a cylinder with a 3 nm layer of fluorophores at the surface.

## I-SceI experiments
For experiments with the I-SceI meganuclease, cells with the p15a-I-SceI plasmid were cultured in M9 minimal medium, loaded in a microfluidic chip and then incubated at 37 °C in M9 medium with glucose (0.4%), RPMI 1640 amino acid supplement (Sigma-Aldrich R7131, 0.05%), carbenicillin (20 µg ml[−1]) and Pluronic F-108 (21 µg ml[−1]). DSBs were induced by switching for 3 min to medium also containing aTc (20 ng ml[−1]) and IPTG (1 mM), and then for three min to medium with only IPTG. The cells were then imaged while repairing and recovering in the initial medium.

## Serial-dilution plating assay
Bacteria cultures were streaked on LA plates from freezer stock and grown overnight at 37 °C. The following day, 5 ml of LB medium was inoculated with a single colony from the overnight plate and cultured for 6 h at 37 °C. Next, tenfold serial dilutions were made in LB and 4 µl of each dilution was plated on a LA plate, or LA plate containing 1 µg ml[−1] of nalidixic acid. Plates were incubated overnight at 37 °C.

## Search model by extended RecA filament
Homology search will be treated as a diffusion-limited reaction with a transport time for the homologous sequences to reach a reaction radius and a probing time once the sequences have met. The probing time for the correct sequence will be negligible compared to the time for getting the homologous sequences in contact, but the overall reaction will be slowed down by the overwhelming number of incorrect sentences that will need to be probed.

To quantify the situations we start with few approximations. Assume that the RecA–ssDNA filament is a thin rod in the centre of the cylindrical nucleoid reaching from pole to pole in the *z* direction, whereas the homologous dsDNA sequence is coiled up at a random position in the nucleoid. The relevant time for the homologous sequences to find each other is the time for a segment in the coiled up dsDNA to diffuse radially into the rod in the centre of the cell. The central realization in our model is that it does not matter at which *z*-coordinate it reaches the rod. This transforms the search problem from 3D to 2D, since we can describe the search process from the perspective of the dsDNA

fragment that is homologous to the ssDNA sequence that happens to be at the $z$-position at which the rod is reached first.

An equivalent way to think about the situation is to consider that the first binding event for many independent searchers (that is, the chromosomal dsDNA segments), that each can bind to one out of many targets (that is, the RecA-bound ssDNA segments), has the same rate as one searcher that can bind all targets. The total rate of binding is $r = \sum_i r_i$, in which $r_i$ is the rate for template segment $i$ to find its homologous ssDNA segment. If we write out the dependance of at which $z$-coordinate, $z_j$, the filament is reached, the total rate can be expressed as $r = \sum_j \sum_i r_i(z_j)p(z_j)$, in which $p(z_j)$ is the probability to reach the filament at position $z_j$ and $r_i(z_j)$ is the conditional rate for segment $i$ of binding given that the filament is reached at $z_j$. Here, the rate of binding is zero unless the template segment matches the ssDNA that is at the specific $z$-position, that is, $r_i(z_{j \neq i}) = 0$ which means that $r = \sum_i r_i(z_i)p(z_i)$, that is, the total binding rate is the same as the binding rate for a single dsDNA segment that can bind at any position at the filament, irrespective of $z$ position, and for which each position is homologous.

## Search time prediction for *E. coli*
In a first-order approximation of how long it takes for a chromosomal dsDNA segment to diffuse from a random radial position in the nucleoid to the filament in its centre, we can use the diffusion limited rate for reaching a rod in the centre of a cylinder[21] of length $2L$, that is, $k = 2\pi(2L)D/\ln(R/r)$, in which $r$ is the reaction radius of the rod, which is assumed to be in the order of a nucleotide (1 nm), and $R$ is the nucleoid radius. The concentration of the searching dsDNA fragment is $c = 1/V = 1/(2L\pi R^2)$, in which $V$ is the nucleoid volume. The average time to reach the rod is therefore $T = V/k = R^2\ln(R/r)/2D$. The nucleoid radius $R \approx 200$ nm and the reaction radius is in the order of $r \approx 1$ nm, although the exact value is inconsequential as only its logarithm enters into the time. The complicated parameter is the diffusion rate constant $D$, as DNA loci movement is subdiffusive and $D$ is therefore lower at a long length scale than a short. The process will however be limited by the long distance movement corresponding to $R$. At the length scale[17] of $R = 200$ nm, $D_R \approx 0.0007$ μm$^2$ s$^{-1}$. The association step of the search process is thus $T \approx (0.2$ μm$)^2 \times \ln(200$ nm/ $1$ nm$)/0.0007$ μm$^2$s$^{-1} = 300$ s = 5 min. If we consider that also the RecA filament is moving on the minute timescale this will only speed up search further.

## Time for probing
The RecA filament will, however, not be accessible for binding all the time since it also needs to probe all other dsDNA segments. If only half of the measured search time (about 10 min) is needed for homologous sequences to meet, 5 min is still available for probing other sequences. Is this sufficient to probe all sequences? If the dsDNA is probed in $n$-bp-long segments, each of the equally long segments of the 2-kb-long ssDNA in the RecA filament will have to interrogate, on average, every $2{,}000/n$ segments of the chromosome. There are 9.6 Mb (4.8 Mb per genome × approximately 2 genomes per cell) of dsDNA in the cell, which corresponds to $9.6 \times 10^6/n$ probing segments. This, in turn, means that each ssDNA segment in the RecA filament needs to test $(9.6 \times 10^6/n)/(2{,}000/n) = 4{,}800$ dsDNA segments. The average time for each test cannot be longer than $300$ s/$4{,}800 \approx 63$ ms, which should be plenty of time, considering that Cas9–sgRNA takes on average 30 ms to perform a similar task[16].

## ATP binding and hydrolysis
ATP-RecA has high affinity towards ssDNA[15] and upon binding results in a stretched and rigid filament with persistence length[16] of about 900 nm. ATP hydrolysis lowers the affinity of RecA to ssDNA and is needed for rapidly discarding mismatched sequences[17,36,37]. ATP turnover is therefore needed to stretch the filament in the intracellular environment where it otherwise would get stuck at partial homologies.

## Alternative models
Alternative models for how to get sequences sufficiently close to probe for homology can come in many different flavours.

The most naive comparison is considering the diffusion limited bi-molecular reaction of a particle with diffusion rate corresponding to the dsDNA segment and a non-diffusive segment of the filament with a reaction radius corresponding to $r$ (about 1 nm) anywhere in nucleiod. Here, the rate of the diffusion limited reaction is $k = 4\pi r D_L$, in which $D_L$ is the diffusion[38,39] rate at the length scale of the cell, whereas the concentration of the segment is the same as above ($c = 1/V = 1/(2L\pi R^2)$). This results in $T = V/k = LR^2/2rD_L$. This should be compared to $R^2\ln(R/r)/2D_R$ in the 2D situation. Importantly, the diffusion rate at the length scale of the nucleoid radius, $D_R$, is one order of magnitude[17] faster than $D_L$. The ratio is $(L/rD_L)/(\ln(R/r)/D_R) \approx 1{,}750$, considering $L = 1$ μm, $R = 300$ nm, $r = 1$ nm and $D_R/D_L \approx 10$. The actual value for $r$ is more important in this case, because the number of rebinding events is more important than in the 2D case.

An intermediate step towards the 2D model is to parallelize the naive model and think of the homologous dsDNA as divided in segments that can search in parallel and independently for their respective homology in the RecA filament. For example, if we divide a 1,750-bases-long filament into segments of 10, the speed would increase by 175 times and the remaining difference compared to the 2D model would be $(D_R/D_L)$ – fold difference of short- and long-range diffusion.

A final model is to consider the homologous DNA as static and that only the RecA filament moves similarly to a knitting needle in a ball of yarn. This situation is not as straightforward to quantify, since we do not know how rigid the RecA filament is on different length scales in the cell. It appears to be flexible on the 100-nm length scale, but the probing interactions will have to occur on the 1-nm scale and we therefore have no knowledge of how fast the filament would explore the genome. It can clearly probe many DNA segments simultaneously[19] but it causes complex constraints to probe sequences with one part of the filament at the same time as the filament should move to explore the rest of the genome. Detailed simulations may be needed to predict the expected search times for this type of model.

## Reporting summary
Further information on research design is available in the Nature Research Reporting Summary linked to this paper.

## Data availability
Raw microscopy data are available at https://doi.org/10.17044/scilifelab.14815802. Source data are provided with this paper.

## Code availability
The computational code used for analysis and plotting is available at https://doi.org/10.17044/scilifelab.14815802.

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

**Acknowledgements** We thank D. Fange, I. Barkefors and D. Jones for helpful input on the manuscript, D. Camsund for the pCRED plasmid, S. Zikrin and P. Karempudi for help with the image analysis. J.W. is an EMBO non-stipendiary fellow (EMBO ALTF 1029-2018). This work was funded by the H2020 Marie Curie Individual Fellowship grant (RecPAIR:842047) to J.W., European Research Council (BIGGER:885360), the Swedish Research Council (2016.06213 and 2018.03958), the Knut and Alice Wallenberg Foundation (2016.0077, 2017.0291 and 2019.0439) and the eSSENCE e-science initiative to J.E., and SSF (FFL15-0031) to I.T.

**Author contributions** J.E., J.W. and A.H.G. conceived the study and interpreted results. A.H.G., J.W. and J.L. conducted the experiments. J.W. and J.L. performed imaging with Cas9-induced breaks; A.H.G. performed imaging with SceI-induced breaks and STED sample preparation. J.W. and A.H.G. wrote analysis code and analysed data, A.H.G., J.W. and P.L. constructed the strains, P.L. made and tested the new RecA fusion, J.E. derived equations and G.C. and I.T. contributed the STED imaging. J.E., A.H.G. and J.W. wrote the manuscript with the input of all authors.

**Competing interests** The authors declare no competing interests.

**Additional information**
**Correspondence and requests for materials** should be addressed to Johan Elf.

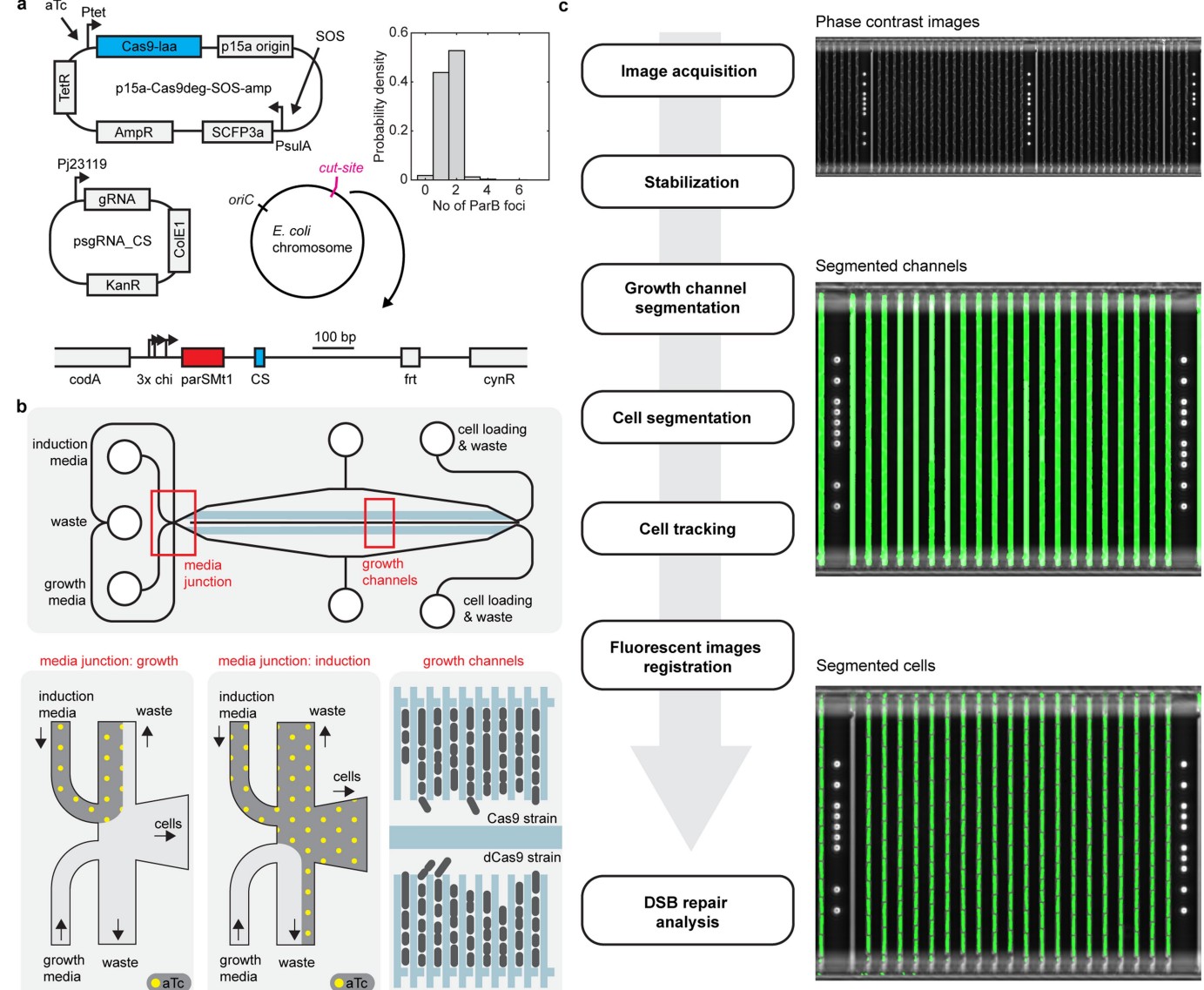

**Extended Data Figure 1 | DSB induction system and image processing.**
**a**, Genetic constructs used in the DSB reporter system. Cas9 and SOS-reporter are encoded on a single plasmid and sgRNA is on a separate plasmid. A map of the circular *E. coli* chromosome with the site at which the cut site was integrated is shown. Bottom cartoon shows a magnified view of the cut site cassette integrated into *codA* locus on the chromosome. The histogram shows the distribution of the number of cut site copies in the cells measured as the number of ParB foci. *n* = 9,263 cells from 3 experiments. **b**, Top, cartoon showing a simplified schematic of a microfluidic chip. Bottom: cartoons showing a medium input junction and flow of the medium during growth (left)

and induction (right) phases of the experiment. The rightmost cartoon shows an arrangement of the growth channels on the chip. Typically strains transformed with an active Cas9 and dCas9 were loaded on separate sides of the chip, to serve as a control for induction strength. **c**, Left: Steps in the image analysis pipeline. Right: *Top* image shows a full field of view (FOV) of a single position of the microfluidic chip. Typically 16 positions were imaged during an experiment. *Middle* image shows an enlarged section of the FOV with an overlaid mask of segmented channels. *Bottom* image shows the same section of FOV as in the middle, but overlaid are segmented cells.

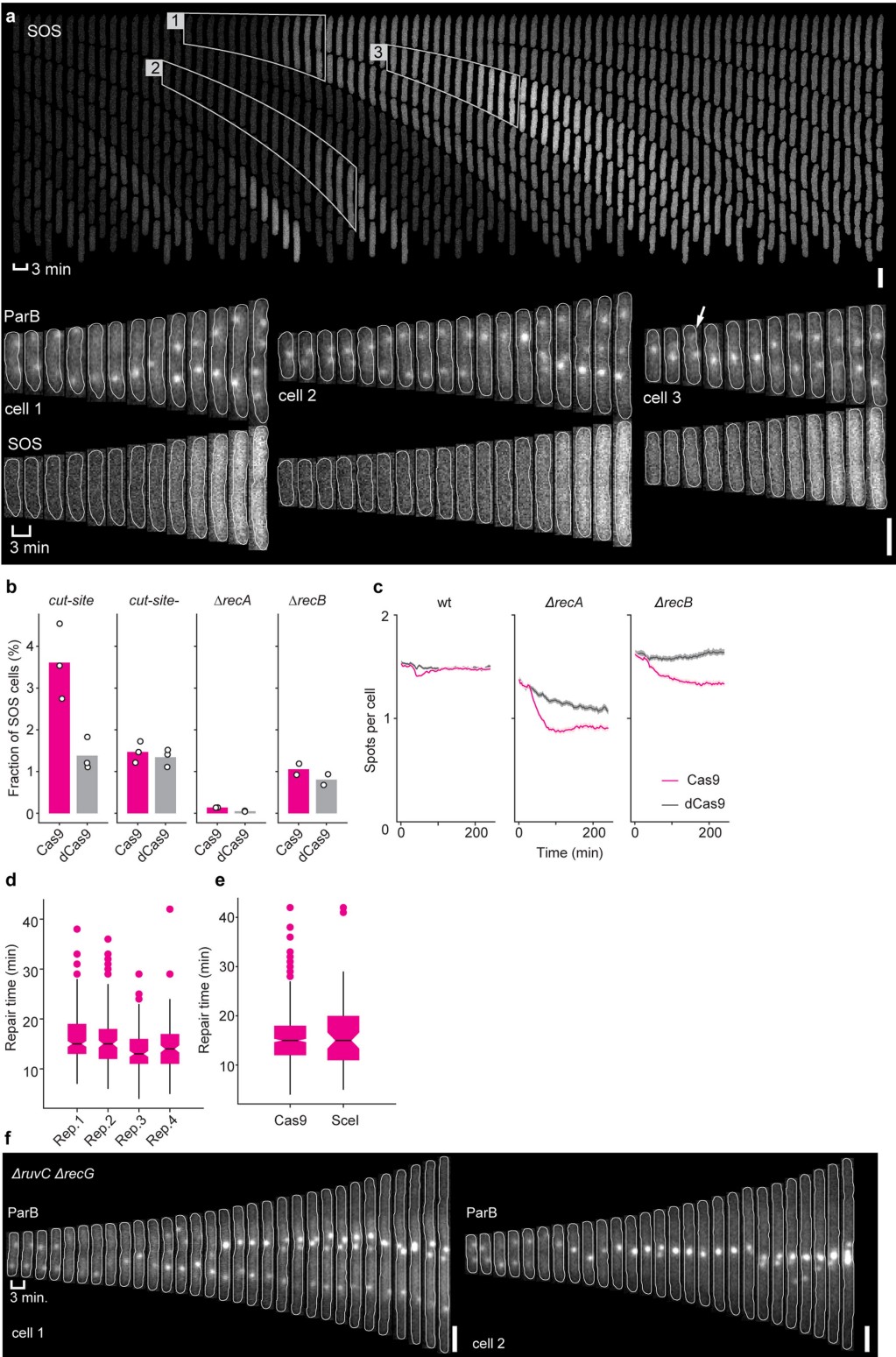

**Extended Data Figure 2 | Specificity of the DSB induction. a**, Top: Cells during a DSB experiment showing SOS reporter channel, cells that activated SOS response are outlined and displayed on the bottom. The arrow points at ParB focus that is lost in the next frame. Scale bar is 2 μm. **b**, Fractions of cells that activated SOS response in different strains and for active, or dead Cas9 variants. Cut-site- strain lacks chromosomal *cut-site*. Bars represent the mean, points represent a mean for each of the biological replicates of the experiments. **c**, Mean ParB foci number per cell after induction of Cas9 or dCas9 (induction was at t = 20 min). Solid line shows mean, light colored area shows 95% confidence interval measured by bootstrap. Wt: n = 3 experiments, *ΔrecA*: n = 2 experiments, *ΔrecB*; n = 2 experiments. **d**, Repair times measured

by ParB focus loss to recovery in each of 4 experimental replicates. Centre line represents the median, box boundaries show 25th and 75th percentiles, whiskers show the most extreme values, or 1.5-fold of upper and lower interquartile range (IQR) if the most extreme values are beyond that cutoff, the solid dots represent the data points beyond the cutoff. N (for each experiment): rep. 1 - 250 cells, rep. 2 - 150 cells, rep. 3 - 174 cells, rep. 4 - 171 cells. **e**, Repair times measured by ParB focus loss to recovery of DSBs caused by either Cas9 (Cas9 data is the same in Fig. 1g) or I-*Sce*I enzyme. Box plots as in d. I-*Sce*I: n = 73 cells, 8 experiments, Cas9: n = 765 cells, 4 experiments. **f**, *ΔruvC ΔrecG* cells after a DSB. Scale bar is 2 μm.

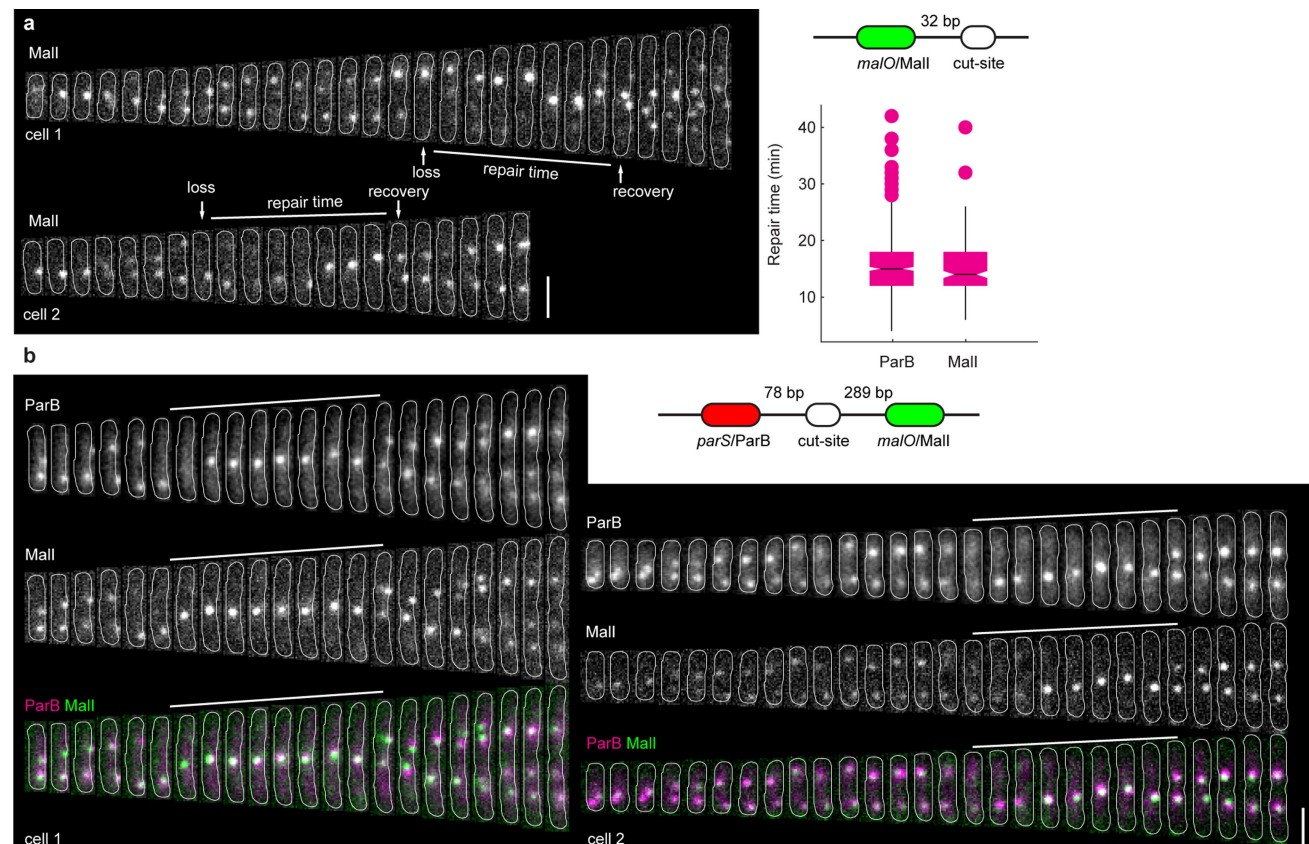

**Extended Data Figure 3 | DSB repair dynamics in presence of different markers. a**, Two example cells undergoing DSB repair. Events of MalI focus loss and recovery are annotated, as well as the repair time. Cell outlines are displayed with a solid gray line. Scale bar is 2 µm. Cartoon shows a cut-site construct with *malO*/MalI marker, the distance between the markers and cut-site is indicated. The box plots display the repair time measurements for the strain with MalI markers compared to the strain with ParB markers (ParB data is the same in Fig. 1g). Centre line represents the median, box boundaries show 25th and 75th percentiles, whiskers show the most extreme values, or 1.5-fold of upper and lower interquartile range (IQR) if the most extreme values are beyond that cutoff, the solid dots represent the data points beyond the cutoff. ParB: n = 765 cells from 4 experiments, MalI: n = 221 cells from 2 experiments. **b**, Two example cells undergoing DSB repair. Repair time is annotated with a solid white line. Cell outlines are displayed with a gray solid line. Scale bar is 2 µm. Cartoon shows a cut-site flanked by *malO*/MalI and *parS*/ParB markers, distance between the markers and the cut-site is indicated.

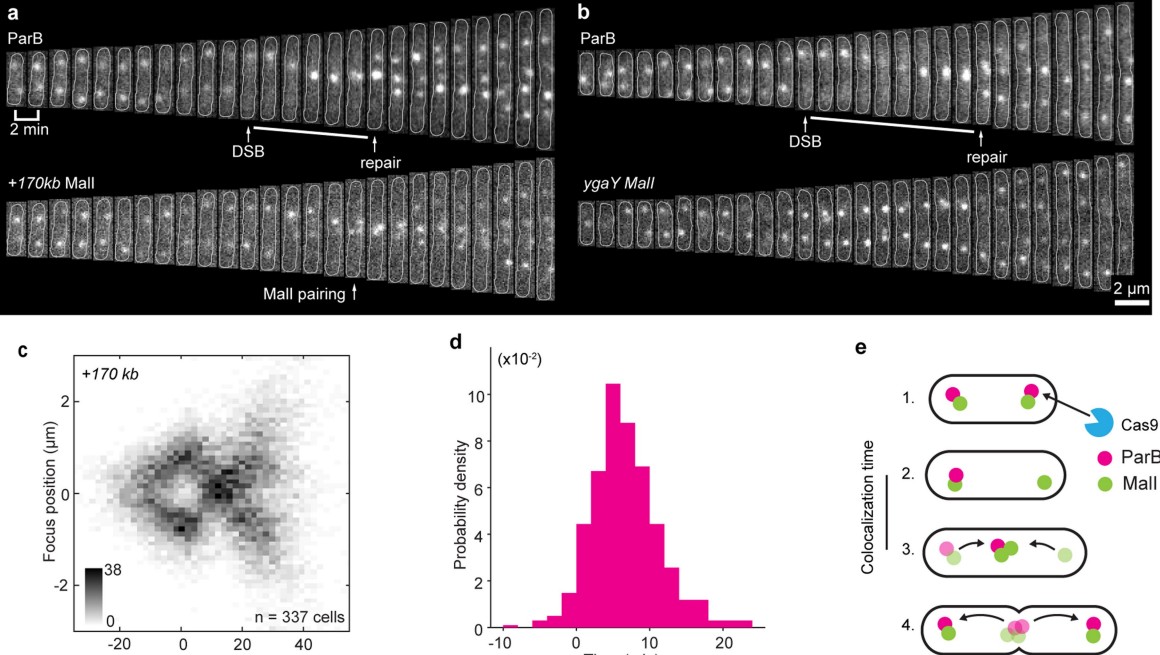

**Extended Data Figure 4 | Paring of the distant homologies. a**, Cell with +170kb MalI markers undergoing DSB repair. ParB focus loss and recovery events are shown. **b**, As in **a**, but for a cell with *ygaY* MalI marker. Event of MalI markers pairing is shown. **c**, Spatial localization of +170kb MalI foci in time in cells undergoing DSB. For each cell, the time was aligned based on the time of ParB focus loss. Color bar displays density of counts, n = 337 cells from 2 experiments. **d**, Time of ParB focus splitting after pairing of the -25 kb MalI foci,

n = 507 cells from 3 experiments. **e**, Cartoon showing events leading to the colocalization of MalI foci in the centre of the cell. 1. DSB induced by Cas9. 2. ParB focus at the site of the DSB is lost due to end resection. 3. Repair template is identified and the homologous loci move and colocalize at the centre of the cell. 4. DSB is repaired, ParB focus is restored, and the homologous loci segregate.

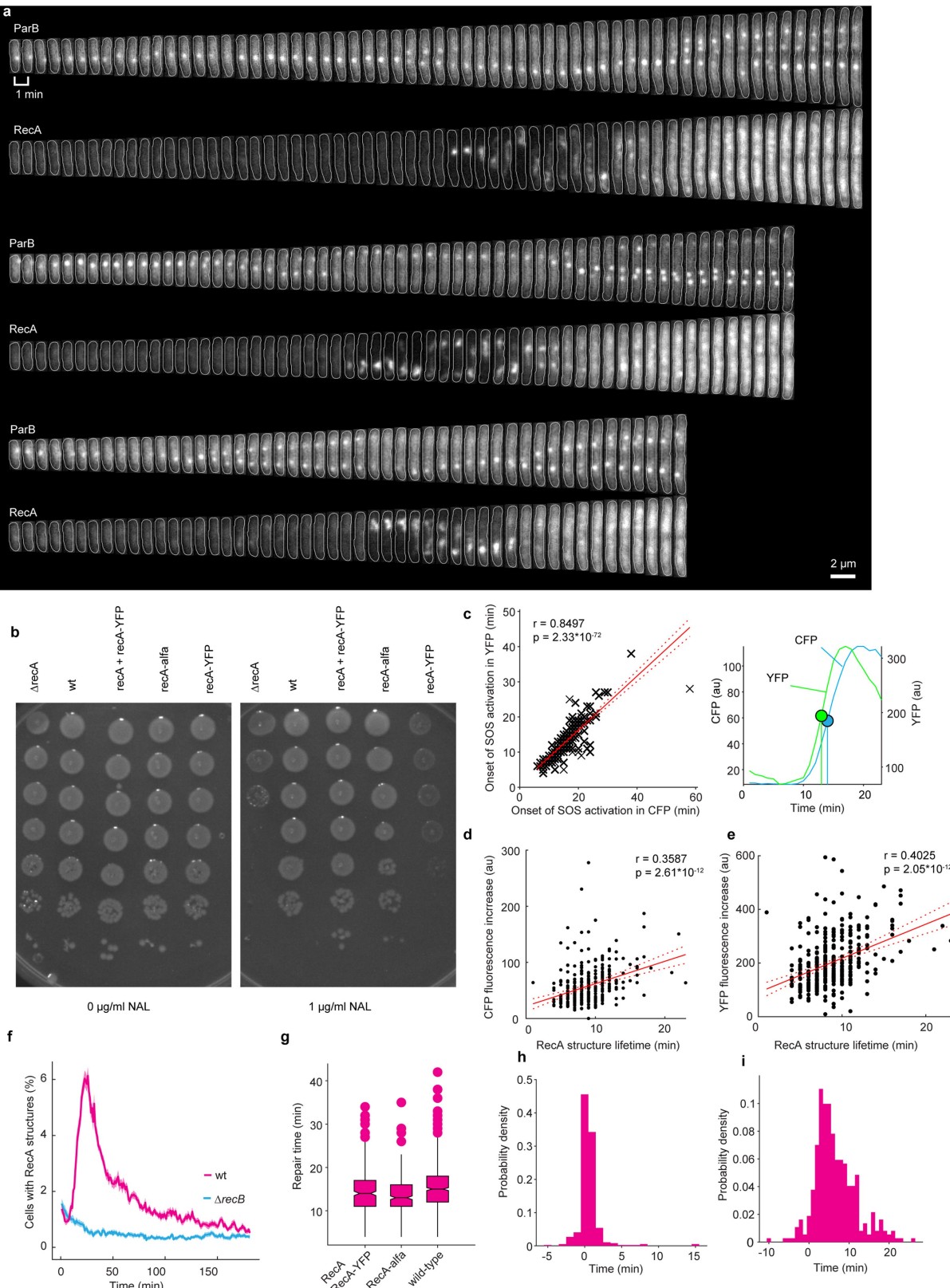

**Extended Data Figure 5** | See next page for caption.

**Extended Data Figure 5 | RecA structures are specific. a**, Cells with ParB and RecA labels undergoing DSB repair. **b**, Growth of strains with different RecA variants without, or in presence of DNA damage induced by nalidixic acid. **c**, Correlation between the time of activation of SOS response measured by fluorescence from plasmid SOS reporter and from RecA expression. SOS activation was measured by the time at which fluorescent signals reached half maximum value. Analysis was restricted only to cells that activated SOS response. Red line shows linear fit to the data (black), dashed lines show 95% confidence interval. Right-hand plot shows SOS (blue) and RecA (green) intensity traces for a single cell. Time of half-maximum signal for each channel is shown. Correlations are Pearson's r, n = 255 cells, from 1 experiment. $P$ values are from F-test. **d**, Correlation between the lifetime of RecA structure and increase of SOS signal intensity for cells undergoing DSB repair. The solid red line shows linear fit, dashed red lines show confidence bounds, correlation is Pearson's r, n = 358 cells from 3 experiments. $P$ values are from F-test. **e**, Same as in **d** but for a correlation between RecA structure lifetime and increase in expression of the RecA-YFP, correlation is Pearson's r, n = 358 cells from 3 experiments. $P$ values are from F-test. **f**, Fraction of cells with RecA structure after the induction of DSB (at time=0), n = 2 experiments. **g**, Repair times measured by ParB focus loss to recovery of DSBs for different RecA variants (wild-type data is the same in Fig. 1g). Centre line represents the median, box boundaries show 25th and 75th percentiles, whiskers show the most extreme values, or 1.5-fold of upper and lower interquartile range (IQR) if the most extreme values are beyond that cutoff, the solid dots represent the data points beyond the cutoff, RecA-YFP: n = 371 cells from 4 experiments, wt: n = 765 cells from 4 experiments, RecA-ALFA: 361 cells from 2 experiments. **h**, Times between appearance of RecA structure to loss of ParB focus, n = 358 cells from 3 experiments. **i**, Times between the disassembly of RecA structure to splitting of ParB focus, n = 358 cells from 3 experiments.

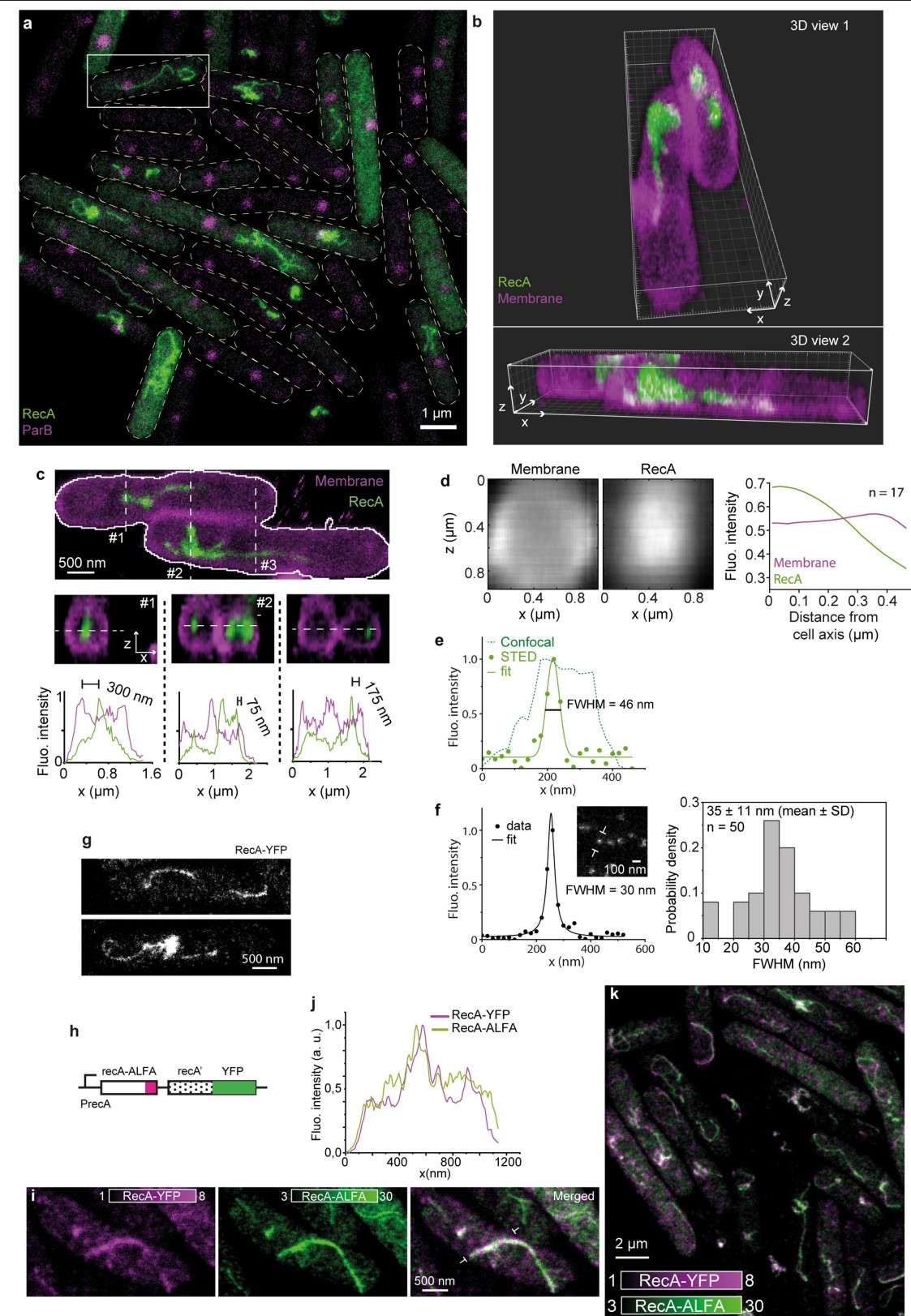

**Extended Data Figure 6** | See next page for caption.

**Extended Data Figure 6 | STED microscopy of RecA filaments. a**, Cells with Cas9-induced DSBs, expressing RecA-ALFA and labeled by anti-ALFA-Star635P antibodies. Star635P was imaged in STED and ParB in confocal microscopy. Cell outlines are indicated by dashed lines. The indicated cell is shown in Fig. 3f. **b**, Volumetric three dimensional view of cells (also presented in Fig. 3i) with extended RecA-ALFA structures, imaged by 3D STED. **c**, Top: 3D STED cross-sections of cells with RecA-ALFA and Nile red membrane stain as indicated in Fig. 3j. Bottom: Line profiles of RecA and membrane intensities as indicated above. Peak-to-peak distances between RecA and membrane are indicated. **d**, Left: Average distribution of membrane dye and RecA in the cross-section of cell segments with extended RecA structures. Right: Radial density of membrane and RecA in the cross-sections to the left, n = 17 cells from 2 experiments. **e**, Line profiles between markers in Fig. 3f, in confocal and STED microscopy. Solid line indicates Gaussian fit of the STED profile. Filament width is measured as the FWHM of the Gaussian fit. **f**, Measurement of STED resolution using Star635P-conjugated antibodies. Left: Profile over single antibody as indicated in the insert, and Lorenzian fit of this profile. Right: Distribution of FWHM of Lorenzian fitted line profiles of antibodies, representing the resolution of the STED microscope, n = 50 from one experiment. **g**, STED image of cells with extended structures and RecA-RecA-YFP construct, labeled by anti-GFP antibody. **h**, Genetic construct expressing both RecA-ALFA and RecA-YFP, inserted in the native *recA* locus. **i**, STED images of cell with extended RecA filament, stained with anti-GFP-Star580 and anti-ALFA-Star635P antibodies. **j**, Fluorescence intensity profile as indicated in **i**, showing overlap between the two antibodies. **k**, Further STED image with immunolabeled cells as in **i**.

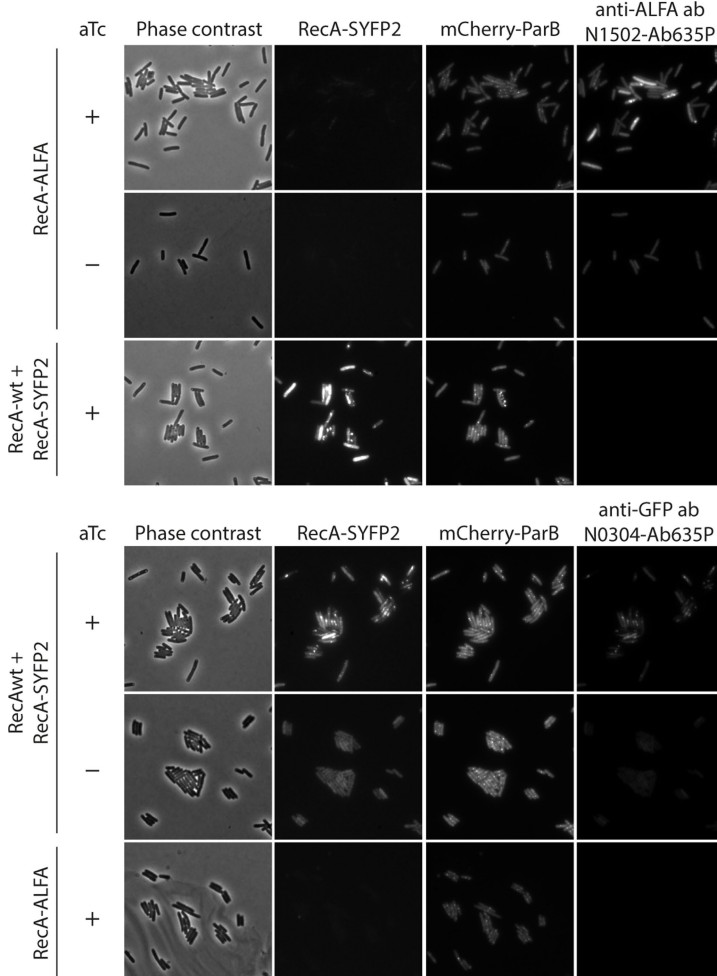

**Extended Data Figure 7 | Validation of antibodies.** Specificity of intracellular antibody labeling, indicated by binding of Star635P-conjugated antibodies specific to ALFA and GFP-derivatives, in strains with or without epitopes, and with and without DNA damage induction.

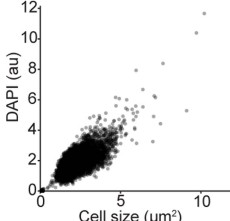

**Extended Data Figure 8 | DNA amount scales with the cell size.** Scatter plot showing DNA content (DAPI) as a function of cell size, n = 9065 cells from 1 experiment.

## Extended Data Table 1 | List of strains used in the study

| strain name | genotype |
|---|---|
| EL666 | ΔrecA::frt-kan-frt |
| EL1171 | MG1655 rph+ ΔmalI::frt intC::P59-malIvenus-frt gtrA::P58-mCherry:parB-SpR codA::lacOsym-lacOCS-parBMt1-frt yahA::malO12x-frt yibU'::PsulA'-scfp3a-KanR |
| EL2504 | MG1655 rph+ lac+ ΔmalI::frt gtrA::P58-mCherry:parB-SpR codA::parBMt1-lacOsym-lacOCS-frt |
| EL2515 | MG1655 rph+ lac+ ΔmalI::frt gtrA::P58-mCherry:parB-SpR codA::parBMt1-lacOsym-lacOCS-frt recA-recX::recAmwg-SYFP2-frt |
| EL2561 | MG1655 rph+ lac+ ΔmalI::frt gtrA::P58-mCherry:parB-SpR codA::parBMt1-lacOsym-lacOCS-frt / p15a-Cas9deg-SOS-amp + psgRNA-CS1 |
| EL2562 | MG1655 rph+ lac+ ΔmalI::frt gtrA::P58-mCherry:parB-SpR codA::parBMt1-lacOsym-lacOCS-frt / p15a-dCas9deg-SOS-amp + psgRNA-CS1 |
| EL2582 | MG1655 rph+ Δlac ΔmalI::frt intC::P59-malIvenus-frt gtrA::P58-mCherry:parB-SpR codA:: malOx12-lacOsym-lacOCS-frt / p15a-Cas9-sos-amp psgRNA-CS1 |
| EL2583 | MG1655 rph+ Δlac ΔmalI::frt intC::P59-malIvenus-frt gtrA::P58-mCherry:parB-SpR codA:: malOx12-lacOsym-lacOCS-frt / p15a-dCas9-sos-amp psgRNA-CS1 |
| EL2618 | MG1655 rph+ lac+ ΔmalI::frt codA::parBMt1-CS1null-lacOsym-lacOCS-frt / p15a-wtCas9-SOS-amp psgRNA_CS1 |
| EL2619 | MG1655 rph+ lac+ ΔmalI::frt codA::parBMt1-CS1null-lacOsym-lacOCS-frt / p15a-dCas9-SOS-amp psgRNA_CS1 |
| EL2621 | MG1655 rph+ lac+ ΔmalI::frt gtrA::P58-mCherry:parB-SpR codA::parBMt1-lacOsym-lacOCS-frt recA-recX::recAmwg-SYFP2-frt / p15a-wtCas9deg-SOS-amp + psgRNA_CS1 |
| EL2622 | MG1655 rph+ lac+ ΔmalI::frt gtrA::P58-mCherry:parB-SpR codA::parBMt1-lacOsym-lacOCS-frt recA-recX::recAmwg-SYFP2-frt / p15a-dCas9deg-SOS-amp + psgRNA_CS1 |
| EL2653 | MG1655 rph+ ΔmalI::frt gtrA::P58-mCherry:parB-SpR codA::parBMt1-lacOsym-lacOCS-frt ΔrecG::frt ΔruvC::frt/ p15a-wtCas9deg-SOS-amp + psgRNA_CS1 |
| EL2672 | MG1655 rph+ lac+ ΔmalI::frt gtrA::P58-mCherry:parB-SpR codA::parBMt1-lacOsym-lacOCS-frt ΔrecA::frt / p15a-wtCas9-deg-amp + psgRNA_CS1 |
| EL2673 | MG1655 rph+ lac+ ΔmalI::frt gtrA::P58-mCherry:parB-SpR codA::parBMt1-lacOsym-lacOCS-frt ΔrecA::frt / p15a-dCas9-deg-amp + psgRNA_CS1 |
| EL2674 | MG1655 rph+ lac+ ΔmalI::frt gtrA::P58-mCherry:parB-SpR codA::parBMt1-lacOsym-lacOCS-frt ΔrecB::frt / p15a-wtCas9-deg-amp + psgRNA_CS1 |
| EL2675 | MG1655 rph+ lac+ ΔmalI::frt gtrA::P58-mCherry:parB-SpR codA::parBMt1-lacOsym-lacOCS-frt ΔrecB::frt / p15a-dCas9-deg-amp + psgRNA_CS1 |
| EL2699 | MG1655 rph+ lac+ ΔmalI::frt gtrA::P58-mCherry:parB-SpR codA::parBMt1-lacOsym-lacOCS-frt recA:ALFA-frt-KmR-frt |
| EL2801 | MG1655 rph+ lac+ ΔmalI::frt codA::parBMt1-lacOsym-lacOCS-frt intC::P59-malIvenus-frt gtrA::P58-mCherry:parB-SpR yahA::malO12-frt-cmr-frt / p15a-wtCas9-SOS psgRNA_CS1 |
| EL2802 | MG1655 rph+ lac+ ΔmalI::frt codA::parBMt1-lacOsym-lacOCS-frt intC::P59-malIvenus-frt gtrA::P58-mCherry:parB-SpR ygaY::malO12-frt-cmr-frt / p15a-wtCas9-SOS psgRNA_CS1 |
| EL2865 | MG1655 rph+ lac+ ΔmalI::frt gtrA::P58-mCherry:parB-SpR codA::parBMt1-lacOsym-lacOCS-frt ybbD::malO12x-frt-Cmr-frt / p15a-wtCas9deg-sos-amp / psgRNA_cs1 |
| EL2834 | MG1655 rph+ lac+ ΔmalI::frt gtrA::P58-mCherry:parB-SpR codA::parBMt1-lacOsym-lacOCS-frt recA-recX::recAmwg-SYFP2-frt DE(recB)::frt / p15a-wtCas9deg-SOS-amp + psgRNA_CS1 |
| EL2837 | MG1655 rph+ recA::ALFA-recAmwg::SYFP2-frt-kan-frt codA::parBMt1-lacOsym-lacOCS-frt / p15a_SceIdeg_amp |
| EL2858 | MG1655 rph+ lac+ ΔmalI::frt gtrA::P58-mCherry:parB-SpR codA::parBMt1-lacOsym-lacOCS-frt recA::syfp2-frt |
| EL2860 | MG1655 rph+ lac+ ΔmalI::frt gtrA::P58-mCherry:parB-SpR codA::parBMt1-lacOsym-lacOCS-frt recA-recX::recAmwg-SYFP2-frt / p15a-dCas9deg-ampR / psgRNA_CS1 |
| EL2861 | MG1655 rph+ lac+ ΔmalI::frt gtrA::P58-mCherry:parB-SpR codA::parBMt1-lacOsym-lacOCS-frt recA:ALFA-frt/ p15a-dCas9deg-ampR / psgRNA_CS1 |
| EL3007 | MG1655 rph+ lac+ ΔmalI::frt intC::P59-malIvenus-frt gtrA::P58-mCherry:parB-SpR codA:: parS-cs-malO-frt / p15a_wtCas9_sos psgRNA_CS1 |
| EL3008 | MG1655 rph+ lac+ ΔmalI::frt intC::P59-malIvenus-frt gtrA::P58-mCherry:parB-SpR codA:: parS-cs-malO-frt / p15a_dCas9_sos psgRNA_CS1 |
| JW2788 (38) | ΔrecB::KmR |
| JW3627 (38) | ΔrecG::KmR |
| JW1852 (38) | ΔrucC::KmR |
| EL1605 (30) | BW25993 DELphi80::yciI INSlacY::ypet INSintC::tetR-dcas9-spR INSaraBAD::T7rnap-tetA DELaraB |
| EL1743 | MG1655 dnaC2 ΔmdoB::aph frt |

**Extended Data Table 2 | List of primers used in the study**

```
cloning primers
```

| | |
|---|---|
| Jwu035 | cttaacagggaagtgagaggtagggtacgggtttttgctgcc |
| Jwu036 | GATATCGAGCTCGCTTGGACgcttggactcctgttgatagatc |
| Jwu037 | GTCCAAGCGAGCTCGATATC |
| Jwu038 | cctctcacttccctgttaagtatc |
| Jwu088 | gatagatccagtaatgacctcag |
| Jwu090 | GTATCGGTCTGATCCTGGGTGCGGCTTACATCCGTTCTCGTGATG |
| Jwu085 | ctgaggtcattactggatctatc |
| Jwu091 | ACCCAGGATCAGACCGATAC |
| Jwu272 | gtaggctgctctacacctagc |
| Jwu273 | tgagtcagctaggaggtgac GCTGCTAACGACGAAAACTACG |
| Jwu274 | gctaggtgtagagcagcctac |
| Jwu263 | cactttacttttatctaatc |
| Jwu264 | gtcacctcctagctgactcaaatc |
| Jwu330 | GCT TCG GCG GGG TTT TTT CGC gttactagtgcttggattctcacc |
| Jwu331 | cttatatcgtatggggctgacttc |
| Jwu332 | tcagccccatacgatataag GCTATGTGCCATCTCGATACTCG |
| Jwu333 | GCGAAAAAACCCCGCCGAAGCGGGGTTTTTTGCGCGTAT tcacaattccttgtacagctc |
| Jwu272 | gtaggctgctctacacctagc |
| Jwu267 | ACTGGCTAATGCACCCAGTA gttttagagctagaaatagcaag |
| Jwu184 | actagtattatacctaggactgag |
| Jwu485 | GAAGAC TA AATC ctgtgcgtttatcgatgctg |
| Jwu486 | GAAGAC TA tcaC TCAGTTAAGCGGCGGC |
| Jwu487 | GAAGAC TA GTGA taaTCTAGCAGGAGGAATTC |
| Jwu488 | GAAGAC TA ACGG Gtcacaattccttgtacagc |
| Jwu489 | GAAGAC TA CCGT CGACCTGCAGTTCGAAG |
| Jwu490 | GAAGAC TA gatt gcaagactttgCTTCC |

```
primers for genomic integrations
```

*malO::-25kb*

| | |
|---|---|
| Jwu056 | GGGTAATGAAAGCAGGTGGTTGATGTAAACCGCTATTCAC cgcgggaattcgattgctag |
| Jwu057 | TGCATCGGGAGAATTAATCGCTGCCAACCTCCCGATGCGC catggtccatatgaatatcctcc |

*malO::+170kh*

| | |
|---|---|
| Jwu103 | GAGGGTGAGTTCCGACCCTGAAACAACAAATAAAATGAACAGTCA cgcgggaattcgattgctag |
| Jwu104 | CAATGTCAGATATCGTAAAAGACCAGTGCAATTCTATGTAAACTC catggtccatatgaatatcctcc |

*malO::ygaY*

| | |
|---|---|
| Jwu520 | AGCAACCGTTTTCGTGGCTTTACACTTATAAGGGTGTTAAGAAG cgcgggaattcgattgctag |
| Jwu521 | AGCAGGCTGTGGAGTAGGGCTTTCCATAGAGTGTACGCTTAACG catggtccatatgaatatcctcc |

*cut-site::codA*

| | |
|---|---|
| lacOsym fw | ACGTAAATACTGGCAGGCGTTTCG |
| cynR-CmR lamda red rev | TCAGGCCTACGAGTTCAGTGCTCTACGTAGGCCGGATAAGGCGTTCACGC GTGTAGGCTGGAGCTG |

**recA-YFP tandem**

| | |
|---|---|
| recA-venus fw | TAGCCGAAACCAACGAGGACTTTGGGAGCATCGTTTCCAAGGGCGAGGAG ctgttcaccgg |
| Dn recA p1 rev | Gcaaaagggccgcagatgcgacccttgtgtatcaaacaagacga TTATTTGGTGTAGGCTGGAGCTGCTTC |

**recA-alfa**

| | |
|---|---|
| RecA-ALFA fw | AGAAACTAACGAAGATTTTGGGAGCATCGTTTCTCGCCTTGAGGAGGAACTGCGCCGCCGCTTAACTGAGtga CCGTCGACCTGCAGTTC |
| Dn recA p1 rev | gcaaaagggccgcagatgcgacccttgtgtatcaaacaagacgaTTATTTGGTGTAGGCTGGAGCTGCTTC |

# nature research

# Reporting Summary

Nature Research wishes to improve the reproducibility of the work that we publish. This form provides structure for consistency and transparency in reporting. For further information on Nature Research policies, see our Editorial Policies and the Editorial Policy Checklist.

## Statistics

For all statistical analyses, confirm that the following items are present in the figure legend, table legend, main text, or Methods section.

| n/a | Confirmed | |
|---|---|---|
| ☐ | ☒ | The exact sample size (*n*) for each experimental group/condition, given as a discrete number and unit of measurement |
| ☒ | ☐ | A statement on whether measurements were taken from distinct samples or whether the same sample was measured repeatedly |
| ☒ | ☐ | The statistical test(s) used AND whether they are one- or two-sided *Only common tests should be described solely by name; describe more complex techniques in the Methods section.* |
| ☒ | ☐ | A description of all covariates tested |
| ☒ | ☐ | A description of any assumptions or corrections, such as tests of normality and adjustment for multiple comparisons |
| ☐ | ☒ | A full description of the statistical parameters including central tendency (e.g. means) or other basic estimates (e.g. regression coefficient) AND variation (e.g. standard deviation) or associated estimates of uncertainty (e.g. confidence intervals) |
| ☐ | ☒ | For null hypothesis testing, the test statistic (e.g. *F*, *t*, *r*) with confidence intervals, effect sizes, degrees of freedom and *P* value noted *Give P values as exact values whenever suitable.* |
| ☒ | ☐ | For Bayesian analysis, information on the choice of priors and Markov chain Monte Carlo settings |
| ☒ | ☐ | For hierarchical and complex designs, identification of the appropriate level for tests and full reporting of outcomes |
| ☐ | ☒ | Estimates of effect sizes (e.g. Cohen's *d*, Pearson's *r*), indicating how they were calculated |

*Our web collection on statistics for biologists contains articles on many of the points above.*

## Software and code

Policy information about availability of computer code

| Data collection | High-throughput microfludic experiments were acquired using Micro-manager 2.0.0. STED microscopy experiments were acquired using ImSpector 0.10 revision 8576 |
|---|---|
| Data analysis | High-throughput microfluidic experiments: Phase image segmentation was done using custom Python (3.7.3) code using Pytorch 1.7.1 library. Cell tracking was done using Baxter algorithm (free software under MIT license). Gramm library (free software under MIT license) was used to create plots.<br><br>STED microscopy: Images were analyzed using ImageJ 1.52p and plots were created with OriginPro2020 |

For manuscripts utilizing custom algorithms or software that are central to the research but not yet described in published literature, software must be made available to editors and reviewers. We strongly encourage code deposition in a community repository (e.g. GitHub). See the Nature Research guidelines for submitting code & software for further information.

## Data

Policy information about availability of data

All manuscripts must include a data availability statement. This statement should provide the following information, where applicable:
- Accession codes, unique identifiers, or web links for publicly available datasets
- A list of figures that have associated raw data
- A description of any restrictions on data availability

Raw microscopy data will be publicly deposited and accession codes will be provided at publication.

# Field-specific reporting

Please select the one below that is the best fit for your research. If you are not sure, read the appropriate sections before making your selection.

☒ Life sciences    ☐ Behavioural & social sciences    ☐ Ecological, evolutionary & environmental sciences

For a reference copy of the document with all sections, see nature.com/documents/nr-reporting-summary-flat.pdf

# Life sciences study design

All studies must disclose on these points even when the disclosure is negative.

| | |
|---|---|
| Sample size | No statistical methods were done to predefine sample size. Sample sizes were chosen based on best practices in the field for experimental methods used. |
| Data exclusions | To measure the dynamics of DSB repair (ParB, MalI, RecA) we analyzed the cells that could be segmented without errors and were tracked for at least 9 minutes. The errors include events when two segmented cells that merge into one, large jumps in size of segmented cells, large movements of the centre of the mass of segmented cell, lost of segmentation between consecutive cells. Next, only the cells that activated the SOS response (by increasing the CFP signal by at least 4-times) were selected for analysis. Finally, only the cells that showed clear DSB repair, that is had 2 separated ParB foci before the DSB that segregated into 2 foci after the repair, were analyzed.<br><br>For 2D STED, cells displaying a single ParB focus and a clear RecA structure were selected. For 3D STED, cells displaying a clear extended RecA structure were selected |
| Replication | All the microfluidic experiments were repeated successfully at least 2 times. For the characterization of the dynamics of the DSB repair we collected at least 300 single cells. The exact number of the repetitions of the experiment and yield in single-cells are stated in the figure captions<br><br>STED microscopy data collection and sample preparation was repeated at least 3 times for each experiment, except the experiment presented in Extended Data Fig. i, k, which was done once. All other experiments were repeated at least twice to ensure reproducibility. |
| Randomization | Not applicable to this work, the study compared a wild-type strain to mutant strains. |
| Blinding | Data analysis was not blinded, however, to reduce potential human bias, the critical steps in the analysis were automated |

# Reporting for specific materials, systems and methods

We require information from authors about some types of materials, experimental systems and methods used in many studies. Here, indicate whether each material, system or method listed is relevant to your study. If you are not sure if a list item applies to your research, read the appropriate section before selecting a response.

## Materials & experimental systems

| n/a | Involved in the study |
|---|---|
| ☐ | ☒ Antibodies |
| ☒ | ☐ Eukaryotic cell lines |
| ☒ | ☐ Palaeontology and archaeology |
| ☒ | ☐ Animals and other organisms |
| ☒ | ☐ Human research participants |
| ☒ | ☐ Clinical data |
| ☒ | ☐ Dual use research of concern |

## Methods

| n/a | Involved in the study |
|---|---|
| ☒ | ☐ ChIP-seq |
| ☒ | ☐ Flow cytometry |
| ☒ | ☐ MRI-based neuroimaging |

## Antibodies

| | |
|---|---|
| Antibodies used | for RecA-ALFA FluoTag-X2 anti-ALFA (N1502-Ab635P) and for RecA-YFP FluoTag-X4 anti-GFP (N0304-Ab635P and N0304-Ab580, NanoTag Biotechnologies) |
| Validation | All antibiodies used in the work are camelid single domain antibodies. The antibodies were used in STED super-resolution experiments. The anti-ALFA antibody N1502 has been shown previously to be highly specific in E. coli as well as HeLa cells (Götzke et. al., 2019, doi: 0.1038/s41467-019-12301-7). Both antibodies were tested in the Elf lab on samples lacking the epitopes, and without DSB damage induction. Our tests confirmed the specificity of the RecA-YFP FluoTag-X4 anti-GFP nanobody for SYFP2 epitope, and FluoTag-X2 anti-ALFA for the ALFA epitope. The validation of the nanobodies is shown in Extended Data Fig. 7. |

