## [Peer Review File · Nature]

Manuscript Title: RecA finds homologous DNA by reduced dimensionality search

Reviewer Comments & Author Rebuttals

Reviewer Reports on the Initial Version:

Referees' comments:

Referee #1:

This is a live-cell investigation on the fundamentally important process of homology recombination (HR), which is accomplished by several proteins including RecA and RecBCD. The details of the homology search is not yet fully understood, and the authors presented the study in live cells to quantitatively measure the underlying processes. The quantitative model to explain the observations is also proposed and discussed.

This is a very impressive set of experiments that are very carefully executed and tested. The approach is very original and elegant. They use a CRISPR system to generate DSB in a controlled way that allows them to quantitatively measure the details of these complex processes.

I like this work very much, and I find these results are truly outstanding. I went through all experimental details, and I believe that almost all possible control experiments have been done. The paper is also very nicely written!

I believe that this paper deserves to be published because of the critical importance of the obtained FULLY QUANTITATIVE results. There are not so many biological studies with quantitative results. But this one is really impressive and well executed, in my opinion.

While I like this work, I still have some comments on the theoretical part. I believe that the authors did not refer to the existing theoretical work that is more general and in one limit is identical to the model proposed here: see Biophys. J. 112:859-867 (2017). In this Biophys. J. paper the model on RecA homology search was introduced. The model was motivated by the in-vitro studies of Kowalczykowski and coworkers in extended flow systems. In this model, it was assumed that the RecA-ssDNA segment is moving parallel to DNA, and occasional associations with DNA are utilized to check the homology. But in the limit of very fast motion of the filament, the model becomes identical to what is discussed in this manuscript. The authors of the Biophys. J. paper solved everything explicitly, and they also estimated times for the homology search. It was found to be several minutes in this limit (coiled DNA), like in these experiments. I recommend that theoretical part in this work to be amended to include the discussions of the theoretical model from Biophys. J., as described above. I am not saying that what was proposed in this work is wrong, but I am saying that something similar was already proposed, and the proper discussion is needed, especially given the fact that the Biophys. J. paper used experimental parameters and obtained search times similar to what is given in this work.

Referee #2:

The manuscript by Wiktor et al. addresses the homology search mechanism of RecA during DSB repair. The study adds significant insight into a fundamental process that has puzzled researchers for a long time. For me, the highlights of the study are: (i) the demonstration that a single repairable DSB has no overall fitness cost because the delay in the cell cycle due to repair is

recovered during subsequent faster divisions; (ii) the lifetime of the RecA filament correlates with SOS induction level; (iii) the precise timing of the series of events during DSB repair; (iv) a feasible model for how the RecA filament rapidly finds a homologous sequence. The following comments are meant to further improve this article.

1. Abstract, "We find that in *E. coli*, DSB repair is completed in 15 minutes without fitness loss": It may be misleading to give such a fixed time of 15 min for DSB repair. For example, a DSB at a replication fork should take less time to be repaired.

2. Can the authors include a control that demonstrates that the ParB/parS system does not perturb the repair process (DSB end resection and homology search)?

3. The chromosomal location of the cut site should be stated in the text since this is crucial for the process of DSB repair by HR. A cut site close to the origin of replication will usually exist in multiple copies, whereas a cut site close the terminus will usually be present at a single copy and hence be unreparable in most cells. In light of this point, the authors should show the distribution of copy numbers of the cut sites.

4. It is interesting that the -25 kb and -170 kb markers become paired, but a distant marker does not show any movement during repair. It would be nice to see over what distance from a DSB pairing occurs, i.e. by measuring a few other *MaiI* markers at increasing distances from the cut site.

5. The sentence "The RecA structures were wiggly and stretched across the cells at a time scale of tens of seconds." could be clarified. Does the "stretching across the cells" refer to the initial growth and elongation of the structure or subsequent movements of the extended structure?

6. The authors write "Our model predicts that the search will be completed within 10 min even if each segment of the RecA filament is occupied by probing non-homologous sequences 50% of the time, allowing for an average of 60 ms per probed sequence." It is difficult to follow this sentence in the main text because the numbers appear out of context (the detailed description of the model is in the supplement only).

7. It has been reported that RecA forms "storage bundles" in undamaged cells (doi: 10.7554/eLife.42761), and other studies have suggested that RecA filaments are associated with the inner cell membrane. Such membrane association would not be ideal for the target search model proposed by the authors as it would hinder the sampling of DNA segments by diffusion. It will be helpful for the field if the authors could comment on any discrepancies between their data and those previously reported RecA structures, and any implications on the target search model.

8. It has also been shown that RecA forms small foci only and no extended filament in the case of a DNA replication-dependent DSB (doi.org/10.1083/jcb.201803020). Would the model presented here need to be revised to be consistent with the results reported by Amarh et al.?

9. I am wondering about the function of RecA induction via the SOS response in light of the results that the homology search is completed within 10 min. RecA expression becomes maximal after 15-20 min when the DSB repair process is already finished. It would hence appear that SOS induction of RecA is of little benefit for the repair of a single DSB?

10. Fig. 3f: It would be clearer to split this image into two panels with green/purple and grey colour scales rather than merged within the same image.

Referee #3:

This is an interesting manuscript, which describes experiments designed to address the unresolved question of homology searching in homologous recombination. A variety of large-scale filamentous structures of RecA protein (often referred to as “bundles”) have been observed by several research groups, and their existence has been proposed to be associated with facilitating homology searching. However, how (or indeed whether) they do this is far from clear. What is clear is that such structures are not required to repair replication-dependent DNA double-strand breaks (DSBs), where the repair template is located close to the broken chromosome, in which situation only RecA foci are observed. In the work by Viktor et al., a thinner, more dynamic form of RecA filament than has been described by other groups is observed in response to a DSB, where the repair template is not expected to lie close by. This structure can be seen over a relatively short time frame (10 min or so) during which chromosomal loci marked with fluorescent tags move towards the centre of the cell to form a single focus that subsequently splits to generate the presumed pair of repaired chromosomal loci. The manuscript proposes an interesting model in which the RecA filament facilitates homology searching by reducing the dimensionality of the search.

Important issues

1. The interpretations of the two main observations in this manuscript seem to contradict each other. In the first observation, foci are seen that are interpreted as movement of both the broken and unbroken homologous sites towards the centre of the cell, where a single focus is formed before splitting to generate two foci presumed to represent the repaired chromosomal locus and the template chromosomal locus. In the second observation, a thin dynamic RecA filament is formed along the length of the cell and this is interpreted as allowing a reduced dimensionality of homology searching as it is interpreted to contain an extended single-stranded region of DNA. This extended single strand stretches the region to be searched across the cell so that, if the template for repair is not similarly extended but is relatively compact and yet dynamic, all that needs to happen is for the template region to move to the filament for regions of homology to come into close proximity. This implies that the template DNA finds its region of homology in the ssDNA without moving along the longitudinal axis of the cell. On the other hand a region of DNA adjacent to the stretched out region of ssDNA might be expected to be pushed towards a cell pole as the ssDNA covers the length of the cell. Instead, what is observed (as mentioned above) is movement of both the region adjacent to the break and the unbroken template towards of the centre of the cell.

How do the authors reconcile the observed longitudinal movement of both broken and unbroken loci towards the centre of the cell with the 2D search of the unbroken template for the RecA filament (that necessarily contains a region of homology anywhere along its length) implying no longitudinal movement during homology searching?

Has the homology search completed before the movement of the foci towards the centre of the cell? If so can the authors discuss what is causing the movement to the centre of the cell? Could this be tension between the recombining chromosomes following release from the RecA filament but before completion of recombination? What is consistent with the microscopy?

The way the text is written leads the reader to think that pairing is occurring in the centre of the cell (e.g. lines 78-79 “Imaging either of the two *malI* markers showed that both sister loci move and pair together in the middle of the cell”). This cannot be where they initially pair if the reduced dimensionality model proposed is correct.

The authors need to present a model that describes their key observations, and movement of the recombining loci to the centre of the cell is one of those key observations.

2. The authors have ignored the fact that a DNA DSB contains two broken ends. If the ssDNA formed by resection at each of these ends forms a RecA-coated filament that is matured into a filament stretching the ssDNA across the cell, one might expect two filaments to be formed for

each DSB. How often are there two filaments in each cell? If there is only one filament, this implies that both regions of ssDNA are in the same filament. If there is one filament containing two regions of ssDNA stretched across the cell, this doubles the chance of the template region finding one or other side of the break and once that is found, ssDNA corresponding to a region of the other side of the break will also be there at the same location in the filament (as long as both regions are stretched across the entire cell). This seems like a nice solution to the issue of finding both ends of the DSB but is not commented on in the modelling. If there are two filaments in different parts of the cell (as appears to be the case in some of the images shown), this causes a problem as there are two regions of homology that both need to find the same unbroken locus but are at different locations.

The modelling needs to take into account the expectation that there are two recombining ends, and this modelling needs to be supported by the number of filaments formed in each cell and their coverage of the length of the cell. The model will not work if filaments only extend across parts of the cell as this could result in sequestration of potentially recombining loci in different parts of the cell.

3. On line 27 it is stated that this is a 2D search. My feeling is that this is more akin to a search in two stretched out dimensions and one curled up dimension. There may be a search in an extended 2D space to find the RecA-ssDNA filament, but then movement in the curled up third dimension of the unbroken target to actually align the correct sequences.

4. There is no explanation of why the break is repairable, except that the authors examined cells with one disappearing ParB focus out of two. How was this established for the markers adjacent to the break that are not part of the resected DNA? Why is Cas9 not cleaving all DNA molecules more often?

5. In Fig. S2a, cell 3 shows an increase in SOS level but no loss of ParB, which should not happen (at least not often enough to be shown as an example). Is this a normal type of behaviour?

6. Figs 2 and S3 present data about the *ygaY* locus on the other side of the chromosome, whereas the main text describes data from the *yahA* locus. This is confusing, especially given that these two genes are not near each other.

7. In line 71, it is stated that there is no fitness cost. This has not been established. To do this, experiments similar to those carried out by Darmon et al. (PLoS One, 2014) would need to be carried out. In that study, it was shown that a replication-dependent break had a cost of 0.6% per generation, and I would expect the cost of repairing a replication-independent break (where homology searching is required) to be greater than this.

8. Fig. 1d is not easy to understand. It might be because it is not well described. It is not clear what is counted and how this relates to the time of half-maximum SOS reporter signal.

9. My understanding is that the cross-sectional diameter of a RecA-ssDNA fibre is of the order of 12.5 nm. How does this translate into a structure with a cross-sectional diameter of 60 nm? In Fig. 4, a RecA-ssDNA filament is depicted, but this cannot be correct. As mentioned above there may be two RecA-ssDNA filaments + more RecA to make up the 60 nm filament. The "more RecA" is a bit problematic as it would only tend to obscure the RecA-ssDNA filaments.

10. For Fig. S5a, the legend is not clear. The reader does not understand what is presented there.

11. The figures are not referred to properly. Often a figure is mentioned when another containing the results is referred to; for example, in line 100 Fig. S3 is referred to when it should be Fig. S4; Fig. S2c is said to give information about *recG* and *ruvC* mutants that are not presented there; some figures are not referred to at all (Fig. 1b for example).

12. The left sides of Figs 3 and S3 have been cut off, removing important information.

13. Fig. 3k: The legend says the image is "immunostained for RecA-YFP" whereas on the figure it is indicated that it is RecA-ALFA.

14. The model assumes that the RecA filament passes through the middle of the cell (and the middle of the nucleoid). However, the filaments observed by Lesterlin et al. (Nature, 2014) were clearly located between the nucleoid and the inner membrane. Can the authors clarify why they have generated a model inconsistent with previous data? Do the authors have data showing that their filaments pass through the middle of the nucleoid? If so, do they believe that the location of the filaments observed by Lesterlin et al. (Nature, 2014) are artefactual? This is important because a model of homology searching with a filament that lies outside the nucleoid is different from a model where the filament passes through the centre of the nucleoid.

Minor issues

15. Line 48: Fig. 1d should have a non-capitalised d.

16. Fig. 1g: There is an extra "i" in density in the y-axis title.

Author Rebuttals to Initial Comments:

Referee #1:

This is a live-cell investigation on the fundamentally important process of homology recombination (HR), which is accomplished by several proteins including RecA and RecBCD. The details of the homology search is not yet fully understood, and the authors presented the study in live cells to quantitatively measure the underlying processes. The quantitative model to explain the observations is also proposed and discussed.

This is a very impressive set of experiments that are very carefully executed and tested. The approach is very original and elegant. They use a CRISPR system to generate DSB in a controlled way that allows them to quantitatively measure the details of these complex processes.

I like this work very much, and I find these results are truly outstanding. I went through all experimental details, and I believe that almost all possible control experiments have been done. The paper is also very nicely written!

I believe that this paper deserves to be published because of the critical importance of the obtained FULLYQUANTITATIVE results. There are not so many biological studies with quantitative results. But this one is really impressive and well executed, in my opinion.

We would like to thank the reviewer for these appreciative words and for highlighting the relevance and quality of the work present in our manuscript. We appreciate that the reviewer noticed and understand the importance of the quantitative description we presented in the manuscript in support of our experimental results.

While I like this work, I still have some comments on the theoretical part. I believe that the authors did not refer to the existing theoretical work that is more general and in one limit is identical to the model proposed here: see Biophys. J. 112:859-867 (2017). In this Biophys. J. paper the model on RecA homology search was introduced. The model was motivated by the in-vitro studies of Kowalczykowski and coworkers in extended flow systems. In this model, it was assumed that the RecA-ssDNA segment

is moving parallel to DNA, and occasional associations with DNA are utilized to check the homology. But in the limit of very fast motion of the filament, the model becomes identical to what is discussed in this manuscript. The authors of the Biophys. J. paper solved everything explicitly, and they also estimated times for the homology search. It was found to be several minutes in this limit (coiled DNA), like in these experiments. I recommend that theoretical part in this work to be amended to include the discussions of the theoretical model from Biophys. J., as described above. I am not saying that what was proposed in this work is wrong, but I am saying that something similar was already proposed, and the proper discussion is needed, especially given the fact that the Biophys. J. paper used experimental parameters and obtained search times similar to what is given in this work.

Thanks for reminding us to re-read the model in Kochugaeva et al (here referred to as 'K-model'). The model is relevant to consider but also differs from ours on a few important points. We will discuss these here:

First, in the K-model, the RecA filament is short compared to the length of the coiled up target DNA. Second, the RecA-ssDNA jumps between different potential target segments, which are the size of the RecA filament, and in the K-model, the time for each jump is taken to be the physical time to diffuse this distance by the RecA filament. In our model, this time is instead set by the radial diffusion of the target DNA to the RecA filament. Third, as a consequence of the first two points, probing in the K-model needs to take place along the whole target DNA sequentially and thus, the time to find the target depends on the length of the target DNA. This is in conflict with our model where search is parallelized.

In the limit of fast diffusion, the K-model gives the search time $T = (L-1)/K_{\text{off}} + L/k_{\text{on}}$, i.e. a dependence on the length of the target DNA L , in units of length of the RecA filament. In our experiments, we do not observe this dependence of the target length. However, if the K-model is applied to the in vivo case where the filament is the length of the cell, this means that $L=1$ and $T=1/k_{\text{on}}$, it predicts the correct result, but does not add much physical insight to the process. Our model can be seen as a way to capture the physics that goes into the k_{on} of the K-model.

Taken together, as far as we can tell, the K-model describes the situation in the Kowalczykowski experiment well, where the target is significantly more extended than the filament and the filament can be modelled as freely diffusion cylinders, but it does not capture the essence of what we see in the cell where $L=1$. We now cite Kochugaeva et al. in the model description. Their results seem correct for the situation that they model.

Referee #2:

The manuscript by Wiktor et al. addresses the homology search mechanism of RecA during DSB repair. The study adds significant insight into a fundamental process that has puzzled researchers for a long time. For me, the highlights of the study are: (i) the demonstration that a single repairable DSB has no overall fitness cost because the delay in the cell cycle due to repair is recovered during subsequent faster divisions; (ii) the lifetime of the RecA filament correlates with SOS induction level; (iii) the precise timing of the series of events during DSB repair; (iv) a feasible model for how the RecA filament rapidly finds a homologous sequence. The following comments are meant to further improve this article.

We would like to thank the reviewer for these kind words and the insightful summary of the manuscript.

1. Abstract, "We find that in E. coli, DSB repair is completed in 15 minutes without fitness loss": It may be misleading to give such a fixed time of 15 min for DSB repair. For example, a DSB at a replication

fork should take less time to be repaired.

We agree with this remark and in the revised abstract we have explicitly pointed out that a DSB between segregated chromosomes is repaired in 15.2 ± 4.99 min (mean \pm SD) minutes. We believe that we now are specific enough so readers will not assume that it also holds true for breaks at the replication fork, such as caused by SbcCD cutting a hairpin structure.

2. Can the authors include a control that demonstrates that the ParB/parS system does not perturb the repair process (DSB end resection and homology search)?

We appreciate the suggestion. We have now redone the measurement of DSB repair time in the strain in which the ParB/parS system was replaced with malO/MalI. The results are in excellent agreement (15.2 ± 4.99 min (mean \pm SD) for ParB vs 15.08 ± 4.74 (mean \pm SD) for MalI), despite the use of different markers. Thus we can assume that the presence of markers is not interfering with the repair as it would be unlikely that two unrelated markers caused exactly the same defect. We have added the data to the Fig. S2.

3. The chromosomal location of the cut site should be stated in the text since this is crucial for the process of DSB repair by HR. A cut site close to the origin of replication will usually exist in multiple copies, whereas a cut site close the terminus will usually be present at a single copy and hence be unrepairable in most cells. In light of this point, the authors should show the distribution of copy numbers of the cut sites.

We thank the reviewer for this accurate remark. We have considered this effect of chromosome position on the copy number in replicating cells and we selected the one that will give between 1 and 2 spots per cell (in our growth condition: 1.5 ± 0.6 spots per cell). The information of chromosomal integration (next to *codA* locus) was shown in Fig. S1 and was stated in the material and methods section. The information about mean spot number per cell was also presented in Fig. S2. We agree with the reviewer that this is important information and we also included the information about the distribution of cut-site numbers in the main text of the revised manuscript. We now also clearly point out that all analysis of repair times and RecA structures in live cells is made on cells that go from two to one, and again to two spots, in order to avoid the error introduced if there are multiple repair templates.

4. It is interesting that the -25 kb and -170 kb markers become paired, but a distant marker does not show any movement during repair. It would be nice to see over what distance from a DSB pairing occurs, i.e. by measuring a few other MalI markers at increasing distances from the cut site.

We agree with the reviewer that it would be interesting to analyse the behaviour of additional chromosomal markers, however, we reason that it would not contribute to better understanding of the search mechanism, which is the main focus of this work. The mid-cell pairing of the fluorescent markers takes place already after the homology is identified and it therefore likely reflects an event that is a consequence, and not the cause of the homologous search. Those additional experiments could be a scope of follow-up work that focuses on the late stages of repair.

5. The sentence "The RecA structures were wiggly and stretched across the cells at a time scale of tens of seconds." could be clarified. Does the "stretching across the cells" refer to the initial growth and elongation of the structure or subsequent movements of the extended structure?

We have now rephrased this sentence and clarified that we refer to the the dynamics of the filament in

its extended form.

6. The authors write “Our model predicts that the search will be completed within 10 min even if each segment of the RecA filament is occupied by probing non-homologous sequences 50% of the time, allowing for an average of 60 ms per probed sequence.” It is difficult to follow this sentence in the main text because the numbers appear out of context (the detailed description of the model is in the supplement only).

In the revised text we describe that it is not just the time for the template to meet the right RecA-ssDNA segment that is important for search but also that it is unoccupied, i.e., not interrogating other potential templates. The model allows an average time for interrogation of up to 60ms, which is longer than the <30ms that we measured for the similar interaction when sgRNA-dCas9 probes dsDNA in vivo (Jones et al. *Science* 2018)

7. It has been reported that RecA forms “storage bundles” in undamaged cells (doi: 10.7554/eLife.42761), and other studies have suggested that RecA filaments are associated with the inner cell membrane. Such membrane association would not be ideal for the target search model proposed by the authors as it would hinder the sampling of DNA segments by diffusion. It will be helpful for the field if the authors could comment on any discrepancies between their data and those previously reported RecA structures, and any implications on the target search model.

In the revised manuscript we address the subcellular location of the RecA filaments. We did not observe the storage clusters reported in Ghodke et al. using our RecA fluorescent fusions (either RecA-RecA-YFP, or RecA-ALFA). in our experiments, but since Ghodke et al. describe the storage clusters as inactive forms of RecA, not bound to ssDNA, the clusters should not influence the mechanism presented in our work. The previous work describing polar bodies was done using a RecA4155-GFP fusion that is only partially active(originally from in DOI: [10.1101/136529](https://doi.org/10.1101/136529)), whereas the constructs used in our work are fully functional. We are concerned that those storage clusters may be attributed to the particular RecA-GFP fusion, not to RecA itself.

To clarify the position of our filaments relative to the membrane, we have now performed 3D-STED microscopy of RecA, together with membrane and DNA staining. (Fig 3h-k, S6c). The images demonstrate that the filaments do not associate with the membrane, but rather typically go through the central region of the cell, through the nucleoid. Notably, the dynamics of ‘our’ filaments are also drastically different from those of bundles. We observe structures with a lifetime of a couple of minutes, whereas the bundles previously reported are stable for tens of minutes up to hours (Fig. 5 in Ghodke et al., Fig. 2 in Lesterlin et al.).

It is intriguing that our filaments behave differently from the ‘bundles’, and we have tried to reconstitute the bundles in our system using our fusions (RecA+RecA-YFP, or RecA-Alfa) but unsuccessfully. However, when the chromosome has suffered unspecific DNA damage (by nalidixic acid or an unspecific restriction enzyme), we observed that multiple RecA filaments arise in single cells, likely reflecting multiple sites of DNA damage, and that those filaments seemed to interact with each other and sometimes appeared to be excluded from the nucleoid area. However, those structures were uniformly thin, thus, lacked the thick central body characteristic to the bundle.

Structures formed by RecA-RecA-YFP construct after treatment with nalidixic acid. Structures have been visualized by binding with anti-GFP single domain antibodies.

Furthermore, with the mutant RecA4155-GFP, we have observed long-lived (>10 min) structures significantly thicker than the thin filaments we observe with RecA+RecA-YFP and RecA-ALFA. These have a thicker central region (average width 124 nm, see figure below), and often thinner extensions (80 nm) at the ends and resemble the bundles previously reported. Such structures were not observed for the RecA fusions used in our manuscript. In the discussion of the revised manuscript, we state that the structures observed in this manuscript are different from the bundles.

Structures displayed by RecA4155-GFP, with width measured by FWHM at the center and at the ends.

8. It has also been shown that RecA forms small foci only and no extended filament in the case of a DNA replication-dependent DSB (doi.org/10.1083/jcb.201803020). Would the model presented here need to be revised to be consistent with the results reported by Amarh et al.?

Following the reviewer's suggestion we now discuss the work of Amarh et al. in the revised manuscript. We believe that the spots observed in Amarh et al. are short, unexpanded filaments. Our imaging shows that the RecA first forms a spot that later extends into the filament and we believe that this spot represents RecA loaded onto ssDNA. The DSB induction system in Amarh et al. (SbcCD

cutting the lagging strand at the replication fork) creates breaks when the homologies are in a close proximity, below the diffraction limit of conventional microscopy, thus there is no need for the RecA-ssDNA to extend into a long filament. Our model intends to explain the situation of fast search between distant homologies, where the extended filament is a critical part. Our model does not make any specific prediction in the case of breaks very close to the template.

9. I am wondering about the function of RecA induction via the SOS response in light of the results that the homology search is completed within 10 min. RecA expression becomes maximal after 15-20 min when the DSB repair process is already finished. It would hence appear that SOS induction of RecA is of little benefit for the repair of a single DSB?

We thank the reviewer for this contemplative question. The RecA induction (as well as the SOS) dynamics in our analyses were measured based on the fluorescent reporter and are affected by the maturation time of the fluorescent protein (for SYFP2 protein fused to RecA it is about 4 minutes, for SCFP3a about 6 minutes). The actual time from the DSB until maximal SOS expression must be shorter but we did not adjust for that in our analysis since we were not confident exactly how long the maturation takes in our conditions.

It is known that the SOS response is required for the DSB repair and is deficient in LexA3 background (which cannot induce SOS response, for example: <https://doi.org/10.1371/journal.pone.0110784>). We also tested whether the induction of SOS is required for the repair of single DSBs in our system and we have repeated the experiment in the lexA3 background. In line with previous work, cells fail to repair DSBs and don't induce the SOS. Unfortunately, we cannot conclude whether the concentration of RecA itself, or another factor within the SOS pathway is limiting the repair.

Our analysis pipeline relies on the SOS response to pick out the relevant cells and here LexA3 mutation prevents us from doing so. Because we cannot process this experiment the same way as other microscopy experiments, we rather not include this analysis in the manuscript.

10. Fig. 3f: It would be clearer to split this image into two panels with green/purple and grey colour scales rather than merged within the same image.

Fig. 3f, now supplementary figure S6b, has been revised for clarity. Uniform color scales are now used, and cell outlines have been added.

Referee #3:

This is an interesting manuscript, which describes experiments designed to address the unresolved question of homology searching in homologous recombination. A variety of large-scale filamentous structures of RecA protein (often referred to as "bundles") have been observed by several research groups, and their existence has been proposed to be associated with facilitating homology searching. However, how (or indeed whether) they do this is far from clear. What is clear is that such structures are not required to repair replication-dependent DNA double-strand breaks (DSBs), where the repair template is located close to the broken chromosome, in which situation only RecA foci are observed. In the work by Viktor et al., a thinner, more dynamic form of RecA filament than has been described by other groups is observed in response to a DSB, where the repair template is not expected to lie close by. This structure can be seen over a relatively short time frame (10 min or so) during which chromosomal loci marked with fluorescent tags move towards the centre of the cell to form a single focus that subsequently splits to generate the presumed pair of repaired chromosomal loci. The manuscript proposes an interesting model in which the RecA filament facilitates homology searching

by reducing the dimensionality of the search.

We would like to thank the reviewer for carefully reading our manuscript and for the appreciative words.

Important issues

1. The interpretations of the two main observations in this manuscript seem to contradict each other. In the first observation, foci are seen that are interpreted as movement of both the broken and unbroken homologous sites towards the centre of the cell, where a single focus is formed before splitting to generate two foci presumed to represent the repaired chromosomal locus and the template chromosomal locus. In the second observation, a thin dynamic RecA filament is formed along the length of the cell and this is interpreted as allowing a reduced dimensionality of homology searching as it is interpreted to contain an extended single-stranded region of DNA. This extended single strand stretches the region to be searched across the cell so that, if the template for repair is not similarly extended but is relatively compact and yet dynamic, all that needs to happen is for the template region to move to the filament for regions of homology to come into close proximity. This implies that the template DNA finds its region of homology in the ssDNA without moving along the longitudinal axis of the cell. On the other hand a region of DNA adjacent to the stretched out region of ssDNA might be expected to be pushed towards a cell pole as the ssDNA covers the length of the cell. Instead, what is observed (as mentioned above) is movement of both the region adjacent to the break and the unbroken template towards the centre of the cell.

We thank the reviewer for making this remark, which we find intellectually engaging. The reviewer captured the sequence of events presented in our work accurately: DSB, RecA-ssDNA filament extension, the mechanism of search by the undamaged template DNA, and the pairing of the labeled chromosomal loci at mid-cell. However, we cannot agree with the statement that we present contradicting observations. We understand that the reviewer is concerned that we do not see the opposing force on the non-resected DNA applied by the extending RecA-ssDNA filament that would push the adjacent DNA in the opposite direction. One important factor that we need to consider when we reflect on this comment is that the marker at the DSB that is not resected during the DSB repair (-25kb *malO*/*Mall* marker) is at a significant distance from the break (too avoid RecBCD end processing), hence is connected to the RecA filament by a long stretch of flexible dsDNA (8µm contour length). If the opposing force exists (which it might very well do), we would not be able to observe it in our system, due to the too large distance between the DSB and the *malO*/*Mall* marker.

How do the authors reconcile the observed longitudinal movement of both broken and unbroken loci towards the centre of the cell with the 2D search of the unbroken template for the RecA filament (that necessarily contains a region of homology anywhere along its length) implying no longitudinal movement during homology searching?

We understand that the reviewer mentions the movement of the un-resected *Mall* markers toward the centre of the cell. In the manuscript, we claim that the movement of those markers to the centre of the cell happens *after* the homology is found. Thus, the movement of the un-resected *Mall* marker is a consequence of successful search (which we discuss in the paragraph 'Pairing between distant homologies', Fig 2 c & d vs. e). Therefore, the movement of the sister loci to the mid-cell does not compromise the search model presented here. We would also like to note that in our model, the longitudinal movement of the repair template is not important to the search time.

Has the homology search completed before the movement of the foci towards the centre of the cell? If so can the authors discuss what is causing the movement to the centre of the cell? Could this be tension between the recombining chromosomes following release from the RecA filament but before completion of recombination? What is consistent with the microscopy?

We agree with the reviewers reasoning that the movement of sister loci to the mid-cell is happening after the search is finished; we now state that in the 2nd to last sentence of the 'Pairing between distant homologies' paragraph, and in the revised Fig. 2 we also present a cartoon describing the movement of sister Mall foci. Our model is consistent with the microscopy; the -25 kb markers colocalize in the midcell 9 minutes after the loss of ParB focus, the RecA filament disassembles 8.3 minutes after the loss of ParB. Thus the movement of Mall happens after the RecA filament disassembly - an event that likely marks the completion of the homology search. We favour a simple explanation for what drives the movement to mid-cell after search: that the loci move to the centre of the cells due a tug-of-war between the segregating chromosomes that were physically connected in the process of recombination. This could also be described as the 'tension between the recombining chromosomes' that the reviewer mentions. However, we do not have data supporting our simple explanation and we rather not discuss it in this manuscript.

The way the text is written leads the reader to think that pairing is occurring in the centre of the cell (e.g. lines 78-79 "Imaging either of the two mall markers showed that both sister loci move and pair together in the middle of the cell"). This cannot be where they initially pair if the reduced dimensionality model proposed is correct.

We thank the reviewer for spotting this. We agree that the wording of that particular sentence could be misleading and we have improved it in the revised manuscript. We have called the merging of the two malO/Mall markers as 'pairing', and we understand this can be confused with the pairing of broken and repaired templates during DSB repair. We agree with the reviewer that according to our model, pairing of the homologies does not happen at the mid-cell cell. We have changed the wording of that sentence and we now refer to that even as 'colocalisation'.

The authors need to present a model that describes their key observations, and movement of the recombining loci to the centre of the cell is one of those key observations.

We agree with the reviewer that this finding deserves more attention in the manuscript. However, we do not think that this mid-cell colocalisation of Mall markers is critical for the search mechanism, as this pairing in the mid-cell happens after the repair template is found. We have added a cartoon and a description of the Mall movement to the mid-cell in Fig. 2g.

2. The authors have ignored the fact that a DNA DSB contains two broken ends. If the ssDNA formed by resection at each of these ends forms a RecA-coated filament that is matured into a filament stretching the ssDNA across the cell, one might expect two filaments to be formed for each DSB. How often are there two filaments in each cell? If there is only one filament, this implies that both regions of ssDNA are in the same filament. If there is one filament containing two regions of ssDNA stretched across the cell, this doubles the chance of the template region finding one or other side of the break and once that is found, ssDNA corresponding to a region of the other side of the break will also be there at the same location in the filament (as long as both regions are stretched across the entire cell). This seems like a nice solution to the issue of finding both ends of the DSB but is not commented on in the modelling. If there are two filaments in different parts of the cell (as appears to be the case in some of the images shown), this causes a problem as there are two regions of homology that both need to find the same unbroken locus but are at different locations.

The modelling needs to take into account the expectation that there are two recombining ends, and this modelling needs to be supported by the number of filaments formed in each cell and their coverage of the length of the cell. The model will not work if filaments only extend across parts of the cell as this

could result in sequestration of potentially recombining loci in different parts of the cell.

This is an excellent comment and to address it properly, we have now made a strain with different color labels on different sides of the cut site. We find that both markers are lost after inducing the break and draw the conclusion that both ends are resected to ssDNA by RecBCD. Given the symmetry in the situation, it seems unreasonable to believe that only one of the ssDNA-ends would bind RecA. As we only observe one filament of ssDNA-RecA, we agree with the reviewer that it has to include both ssDNA ends, although we can't think of a direct experiment to prove it. The width of the filaments, as determined by STED, cannot exclude either single or double RecA-ssDNA filaments (see answer to question 9). Long, thin filaments also sometimes make a turn at the cell pole and bind back to itself, suggesting a possibility that multiple ssDNA-RecA filaments like to stick to each other. As the reviewer comments, having both the strands in the same filament solves the problem of "finding the other end" once the homology has been located. Furthermore, since the time point for degrading the single RecA filament and the pairing of the loci at mid cell coincide, there is also no time for a second search phase.

3. On line 27 it is stated that this is a 2D search. My feeling is that this is more akin to a search in two stretched out dimensions and one curled up dimension. There may be a search in an extended 2D space to find the RecA-ssDNA filament, but then movement in the curled up third dimension of the unbroken target to actually align the correct sequences.

This is generally the case for diffusion limited reaction. There is a first phase of bringing the reactants together and a second binding phase when the reactants reach the reaction boundary, including for example macromolecules tumbling around each other. The dimensionality of the first phase is normally seen as the dimensionality of the reaction. The interesting thing with 2D reactions, such as on membranes or in this case to reach a rod in 3D, is that the probability of binding once you have reached the reaction boundary is very high because the probability of returning to the same point in 2D is very high whereas it is very low in 3D. This means that the exact movement in the "curled" up dimension is not very important. Furthermore, the subdiffusive nature of chromosomal DNA segments is further improving the sampling in the last step, since diffusion is fast on a short length scale. In summary, we agree with the reviewers assessment that the process is 2D plus a curled up third dimension, and this is implicit in the modelling framework.

4. There is no explanation of why the break is repairable, except that the authors examined cells with one disappearing ParB focus out of two. How was this established for the markers adjacent to the break that are not part of the resected DNA? Why is Cas9 not cleaving all DNA molecules more often?

For clarity we choose to divide this comment into 3 parts and answer them separately:

"There is no explanation of why the break is repairable, except that the authors examined cells with one disappearing ParB focus out of two."

We concluded that the breaks we observe are repairable based on two lines of reasoning. First, when we removed the components of recombination machinery we observed no recovery of ParB foci (Fig. S2c). This shows that the events of ParB loss and recovery is due to activity of homologous recombination, thus the breaks are repaired, which makes them repairable. We have also observed that when there was no homologous template present, e.i. when Cas9 cut all of the cut-sites, the repair took place.

"How was this established for the markers adjacent to the break that are not part of the resected DNA?"

We understand that the reviewer is concerned that some proportion of DSBs could be processed and repaired without the loss of underlying ParB focus. We believe that this cannot be the case in *E. coli* and that the ParB foci will always be ejected after the DSB, at least for constructs similar to ours. Previous work coming from (Wiktor et al., 2018 *Nucleic acid res* <https://doi.org/10.1093/nar/gkx1290>) presented evidence that the loss of ParB markers next to the break is tightly linked to a formation of DSB at a distance of ~100 bp to the break (similar to the DSBs in our work), and at high level of DSB induction all ParB markers can disappear. We have also tested it in our system by a full induction of DSBs with I-SceI enzyme. We reasoned that if all ParB are lost due to resection of DSBs, then cutting off all possible sites would lead to no more foci in cells. This is in fact what we observed (figure below). This experiment cannot be reproduced with Cas9 due to its slow search kinetics preventing it from cutting all the sites simultaneously.

Mean number of ParB foci spots per cell after the high level of induction of I-SceI nuclease starting at time = 0. Cells had a SceI recognition sequence in proximity to the parS site.

“Why is Cas9 not cleaving all DNA molecules more often?”

We have used a low induction protocol in which most cells do not induce any DSBs, and small fractions induce only a single DSB. Another benefit of using Cas9 for this type of work is the slow search (Jones et al, 2017 *Science*), which also makes it more unlikely that all copies of the cut-site are recognised at the same time. We have observed a higher fraction of ‘double cutters’ when we used the I-SceI enzyme before. Still, we have sporadically observed cells with more than 1 DSB, but we did not include them in the analysis because they did not finish the repair, or were difficult to annotate.

5. In Fig. S2a, cell 3 shows an increase in SOS level but no loss of ParB, which should not happen (at least not often enough to be shown as an example). Is this a normal type of behaviour?

No, this is not a normal type of behaviour. The cell number 3 from Fig. S2a does in fact display a loss of ParB focus, the top focus is lost between the frame 3 and 4, but we acknowledge that the quality of that particular focus is lower than the other ones. This is likely due to the ParB marker being slightly out of focus, and reflects the limitation of the depth of imaging with a 100x objective with high NA. We have adjusted the brightness of the displayed image to make the identification of ParB focus loss easier.

However, we do observe a small and constant fraction (~1%) of SOS-induced cells also without Cas9 induction, likely due to endogenous damage during the DNA replication. Those fractions can be seen on Fig. 1d, or Fig. S2b. Similar spontaneous SOS induction was also observed before: <https://doi.org/10.1038/ng2051>.

6. Figs 2 and S3 present data about the ygaY locus on the other side of the chromosome, whereas the main text describes data from the yahA locus. This is confusing, especially given that these two genes

are not near each other.

We are thankful to the reviewer for finding this error, we somehow missed it ourselves. We have erroneously labeled that marker in the text; the main text should refer to the *ygaY* locus and we have corrected it in the revised manuscript.

7. In line 71, it is stated that there is no fitness cost. This has not been established. To do this, experiments similar to those carried out by Darmon et al. (PLoS One, 2014) would need to be carried out. In that study, it was shown that a replication-dependent break had a cost of 0.6% per generation, and I would expect the cost of repairing a replication-independent break (where homology searching is required) to be greater than this.

We thank the reviewer for introducing us to work presented in Darmon et al. The Cas9 induction system unfortunately is not suited to carry out long term competition assay with the constant expression of nuclease. In Darmon et al., the SbcCD nuclease was used to induce the DSB only at the lagging strand of the replication fork, therefore a repair template should always be present. Constant induction of Cas9 leads to a significant proportion of cells with cuts in all sister chromosomes and subsequent cell death, which prevents us from using competition assays with the Cas9 nuclease. We reanalysed and replotted (Fig. 1f) the data and in the revised manuscript we rephrase the claims. The new analysis shows that the mean growth rate is reduced by 4% in the cells undergoing repair of a single DSB, however, this difference is in the order of accuracy for measuring the growth rates in our system.

8. Fig. 1d is not easy to understand. It might be because it is not well described. It is not clear what is counted and how this relates to the time of half-maximum SOS reporter signal.

We have expanded the description of how the time of half-maximum SOS signal is calculated in the figurecaption.

9. My understanding is that the cross-sectional diameter of a RecA-ssDNA fibre is of the order of 12.5 nm. How does this translate into a structure with a cross-sectional diameter of 60 nm? In Fig. 4, a RecA-ssDNA filament is depicted, but this cannot be correct. As mentioned above there may be two RecA-ssDNA filaments + more RecA to make up the 60 nm filament. The "more RecA" is a bit problematic as it would only tend to obscure the RecA-ssDNA filaments.

The observed 60 nm is a result of a convolution with PFS of the imaging system (a Lorentzian $w = 35 \pm 11$ nm). In the main text we now clarify this with the sentence "We estimated the diameter of the filaments to be 37.5 ± 23.5 nm as a deconvolution of the observed 60 ± 13 nm FWHM width (Fig. 3h) with the PFS of the imaging system (Fig. S6d,e)." From STED imaging we can't say if this includes one or two ssDNA-RecA with bound antibodies.

We have updated Figure 4, in the legend we now write that we depict a single filament for simplicity of presentation, however, there must be another filament originating from the second end of the DSB.

'Reduced dimensionality' explains the first contact between the repair template and the ssDNA and is not affected by the presence of the second filament.

10. For Fig. S5a, the legend is not clear. The reader does not understand what is presented there.

We have now clarified that this figure demonstrates the absence of antibody binding when no epitopes are present.

11. The figures are not referred to properly. Often a figure is mentioned when another containing the results is referred to; for example, in line 100 Fig. S3 is referred to when it should be Fig. S4; Fig. S2c is said to give information about recG and ruvC mutants that are not presented there; some figures are not referred to at all (Fig. 1b for example).

This should now have been fixed.

12. The left sides of Figs 3 and S3 have been cut off, removing important information.

We thank the reviewer for pointing this out, it was a mistake that arose during the export to PDF. We have corrected this in the revised manuscript.

13. Fig. 3k: The legend says the image is “immunostained for RecA-YFP” whereas on the figure it is indicated that it is RecA-ALFA.

We thank the reviewer for pointing this out. The annotation in fig 3k, now supplementary figure S6f, has been corrected.

14. The model assumes that the RecA filament passes through the middle of the cell (and the middle of the nucleoid). However, the filaments observed by Lesterlin et al. (Nature, 2014) were clearly located between the nucleoid and the inner membrane. Can the authors clarify why they have generated a model inconsistent with previous data? Do the authors have data showing that their filaments pass through the middle of the nucleoid? If so, do they believe that the location of the filaments observed by Lesterlin et al. (Nature, 2014) are artefactual? This is important because a model of homology searching with a filament that lies outside the nucleoid is different from a model where the filament passes through the centre of the nucleoid.

We agree with the reviewer’s comment that our model describes best a situation where the filament passes through the middle of the cell and is not associated with the inner membrane. We have now performed 3D-STED microscopy of RecA, together with membrane and DNA staining that show that the thin filaments formed during repair of a Cas9 induced DSB are passing through nucleoid (**Fig 3h-k, S6c**). In the discussion in the revised manuscript, we also highlight the differences (the lack of thick central body, fast dynamics, short lifetime) between our structures and the bundles and we explain why we think that the bundles are not involved in the DSB repair.

It is intriguing that our filaments behave differently from the ‘bundles’, and we have tried to reconstitute the bundles in our system using our fusions (RecA+RecA-YFP, or RecA-Alfa) but unsuccessfully. However, when the chromosome has suffered unspecific DNA damage (by nalidixic acid or an unspecific restriction enzyme), we observed that multiple RecA filaments arise in single cells, likely reflecting multiple sites of DNA damage, and that those filaments seemed to interact with each other and sometimes appeared to be excluded from the nucleoid area. However, those structures were uniformly thin, thus, lacked the thick central body characteristic to the bundle.

Structures formed by RecA-RecA-YFP construct after treatment with nalidixic acid. Structures have been visualized by binding with anti-GFP single domain antibodies.

Furthermore, with the mutant RecA4155-GFP, we have observed long-lived (>10 min) structures significantly thicker than the thin filaments we observe with RecA+RecA-YFP and RecA-ALFA. These have a thicker central region (average width 124 nm, see figure below), and often thinner extensions (80 nm) at the ends and resemble the bundles previously reported. Such structures were not observed for the RecA fusions used in our manuscript. In the discussion of the revised manuscript, we state that the structures observed in this manuscript are different from the bundles.

Structures displayed by RecA4155-GFP, with width measured by FWHM at the center and at the ends.

Minor issues

15. Line 48: Fig. 1d should have a non-capitalised d.

We have corrected that in the revised manuscript

16. Fig. 1g: There is an extra “i” in density in the y-axis title.

We have corrected that in the revised manuscript

Reviewer Reports on the First Revision:

Referees' comments:

Referee #1:

I am satisfied with authors' responses to my questions and to questions of other reviewers. I think the paper is significantly improved and I support the publication in the current form.

Referee #2:

I am satisfied with the authors' revision of the manuscript and recommend publication.
Stephan Uphoff

Referee #3:

The authors have satisfactorily addressed my comments. The study is of general interest, presenting a model for homology searching in bacterial cells that is supported by high quality experimental data. Acknowledging that there are indeed two ends at the breaks studied now fills in a previously missing part of the story, and explaining that the movement of loci towards the centre of the cell likely happens after completion of homology searching greatly clarifies the relationship of the model to the data.

Congratulations on excellent work that moves the field forward.
David Leach